# Diffusion Adaptive Text Embedding for Text-to-Image Diffusion Models

**Byeonghu Na**[1]   **Minsang Park**[1]   **Gyuwon Sim**[1]   **Donghyeok Shin**[1]
**HeeSun Bae**[1]   **Mina Kang**[1]   **Se Jung Kwon**[2]   **Wanmo Kang**[1]   **Il-Chul Moon**[1,3]
[1]KAIST, [2]NAVER Cloud, [3]summary.ai
{byeonghu.na,pagemu,gkwlaks4886,tlsehdgur0,cat2507,kasong13}@kaist.ac.kr,
sejung.kwon@navercorp.com , {wanmo.kang,icmoon}@kaist.ac.kr

## Abstract

Text-to-image diffusion models rely on text embeddings from a pre-trained text encoder, but these embeddings remain fixed across all diffusion timesteps, limiting their adaptability to the generative process. We propose Diffusion Adaptive Text Embedding (DATE), which dynamically updates text embeddings at each diffusion timestep based on intermediate perturbed data. We formulate an optimization problem and derive an update rule that refines the text embeddings at each sampling step to improve alignment and preference between the mean predicted image and the text. This allows DATE to dynamically adapts the text conditions to the reverse-diffused images throughout diffusion sampling without requiring additional model training. Through theoretical analysis and empirical results, we show that DATE maintains the generative capability of the model while providing superior text-image alignment over fixed text embeddings across various tasks, including multi-concept generation and text-guided image editing. Our code is available at https://github.com/aailab-kaist/DATE.

## 1 Introduction

Text-to-image generation has recently received significant attention due to its capability to generate realistic and semantically accurate images from textual prompts. This progress has been largely driven by diffusion models [18, 53], particularly with large-scale models such as DALL·E [45] and Stable Diffusion [47]. These models use pre-trained text encoders like CLIP [42] and T5 [43] to encode prompts into embeddings, providing crucial semantic information to diffusion models. Notably, the quality and semantic alignment of the generated images heavily depend on these embeddings [48].

Despite their success, pre-trained diffusion models often struggle with semantic alignment and human preferences. Recent studies have addressed this using external reward functions, either through preference fine-tuning [3, 31, 56] or through applying guidance directly to denoised images during sampling [2]. However, these methods focus on model parameters or intermediate latent variables and overlook text embeddings. Most text-to-image diffusion models use fixed text embeddings throughout the sampling process (upper part of Figure 1a), limiting their adaptability to the evolving generation process. Since different diffusion timesteps influence generation in different ways [6, 59], static embeddings can fail to capture evolving semantics, leading to suboptimal text-image alignment.

To address this limitation, we propose Diffusion Adaptive Text Embedding (DATE), which dynamically updates text embeddings at each diffusion sampling step based on the current denoised image (lower part of Figure 1a). By continuously tuning the embeddings to maximize alignment between the text prompt and the mean predicted image, DATE captures evolving semantics without extra model training or architectural changes. Notably, DATE operates entirely at test time by simply inserting embedding updates into existing sampling procedures. Our theoretical and empirical results demon-

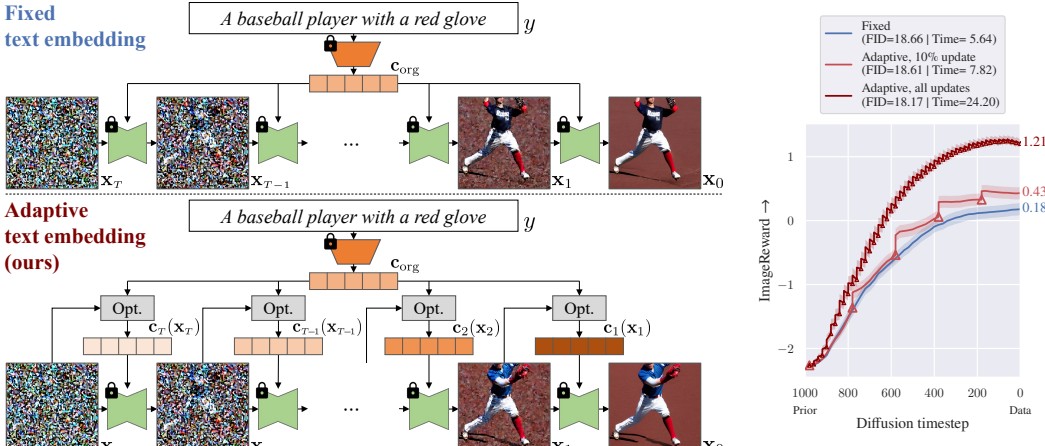

(a) Comparison of fixed and adaptive text embeddings in diffusion sampling

(b) ImageReward for mean predicted images across timesteps

Figure 1: (a) Overview of the conventional fixed text embedding and the proposed adaptive text embedding during the text-to-image diffusion sampling process. Green shapes represent the diffusion model network, orange shapes represent the text encoder, and gray boxes labeled *Opt.* indicate our text embedding optimization, detailed in Figure 3. (b) ImageReward [57], a text-to-image generation metric, for mean predicted images. Red triangles mark the timesteps where text embedding is updated.

strate that DATE improves text-image alignment while preserving the model's original generative capabilities. When evaluated across various diffusion models and samplers, DATE outperforms fixed text embeddings consistently, indicating its agnostic characteristics to both models and samplers. Furthermore, DATE can be effectively integrated into various downstream tasks, such as multi-concept generation and text-guided image editing, highlighting its broad applicability.

## 2 Preliminaries

### 2.1 Diffusion models

Diffusion models consist of two diffusion processes: a forward process and its corresponding reverse process [18]. The forward process is typically defined as a fixed Markov noise process, which perturbs the data instance $\mathbf{x}_0 \sim q(\mathbf{x}_0)$ by adding Gaussian noise:

$$q(\mathbf{x}_{1:T}|\mathbf{x}_0) \coloneqq \prod_{t=1}^{T} q(\mathbf{x}_t|\mathbf{x}_{t-1}), \text{ where } q(\mathbf{x}_t|\mathbf{x}_{t-1}) \coloneqq \mathcal{N}(\mathbf{x}_t; \sqrt{1-\beta_t}\mathbf{x}_{t-1}, \beta_t\mathbf{I}). \quad (1)$$

Here, $\mathbf{x}_{1:T}$ are latent variables for perturbed data, and $\beta_t$ is the variance schedule parameter. Diffusion models aim to approximate the reverse process via a trainable Markov chain with Gaussian transitions:

$$p_{\boldsymbol{\theta}}(\mathbf{x}_{0:T}) \coloneqq p_T(\mathbf{x}_T)\prod_{t=1}^{T} p_{\boldsymbol{\theta}}(\mathbf{x}_{t-1}|\mathbf{x}_t), \text{ where } p_{\boldsymbol{\theta}}(\mathbf{x}_{t-1}|\mathbf{x}_t) \coloneqq \mathcal{N}(\mathbf{x}_{t-1}; \boldsymbol{\mu}_{\boldsymbol{\theta}}(\mathbf{x}_t, t), \sigma_t^2\mathbf{I}), \quad (2)$$

$\boldsymbol{\mu}_{\boldsymbol{\theta}}(\mathbf{x}_t, t)$ is the parametrized mean, $\sigma_t^2$ is the time-dependent variance, and $p_T$ is the prior distribution.

The mean function $\boldsymbol{\mu}_{\boldsymbol{\theta}}(\mathbf{x}_t, t)$ is trained by minimizing the upper bound of the negative log-likelihood [18]. Notably, this mean function can be equivalently expressed, up to a constant, as a score network $\mathbf{s}_{\boldsymbol{\theta}}(\mathbf{x}_t, t)$ that approximates the score function $\nabla_{\mathbf{x}_t} \log q_t(\mathbf{x}_t)$ [53]:

$$\min_{\boldsymbol{\theta}} \mathbb{E}_t \mathbb{E}_{\mathbf{x}_t} \left[ ||\mathbf{s}_{\boldsymbol{\theta}}(\mathbf{x}_t, t) - \nabla_{\mathbf{x}_t} \log q_t(\mathbf{x}_t)||_2^2 \right]. \quad (3)$$

Once the transition kernel $p_{\boldsymbol{\theta}}$ is trained, we sample iteratively from $T$ to near-$0$ using Eq. (2).

### 2.2 Text-to-image diffusion models

Text-to-image generation aims to produce high-quality images that are semantically aligned with a given textual description. Recent advances in diffusion models have greatly improved this task [39, 44, 47]. Text-to-image diffusion models can be formulated as learning a score network with an additional text-conditional input $\mathbf{c}$ to approximate a conditional score function:

$$\min_{\boldsymbol{\theta}} \mathbb{E}_t \mathbb{E}_{\mathbf{x}_t} \left[ ||\mathbf{s}_{\boldsymbol{\theta}}(\mathbf{x}_t, \mathbf{c}, t) - \nabla_{\mathbf{x}_t} \log q_t(\mathbf{x}_t|y)||_2^2 \right]. \quad (4)$$

Here, $\mathbf{c}$ is the text embedding of the text prompt $y$, typically obtained from a pre-trained text encoder [11, 33, 48, 61, 62].

Despite their impressive realism, pre-trained diffusion models often struggle to maintain precise semantic alignment or satisfy human preferences. As the formulation of conditional score network suggests, text-conditional diffusion models can be improved by targeting three components: model parameters $\boldsymbol{\theta}$, perturbed data $\mathbf{x}_t$, and text embedding $\mathbf{c}$. Improvements in each component offer complementary benefits for overall quality and text-image consistency.

**Fine-tuning and data-space guidance**  Most prior works focus on optimizing the model parameters $\boldsymbol{\theta}$ through fine-tuning [3, 12, 24, 31, 56]. These approaches adjust diffusion models using additional curated datasets or reward signals to improve alignment or human preference satisfaction. However, they require extensive retraining and substantial computational cost. Another direction modifies the perturbed data $\mathbf{x}_t$ via external guidance functions. Classifier Guidance [9] steers samples using gradients of time-dependent classifier, while Universal Guidance [2] approximates this approach with a time-independent classifier. While effective, such methods require careful guidance scaling across timesteps, and the guidance component needs to be expressed as a classification probability.

**Prompt optimization**  Another line of work focuses on the text-conditioning component of diffusion models. Prompt-level optimization targets the input text $y$ to produce better conditioning signals [15, 36]. These methods use reinforcement learning with external reward models to train language models that generate refined prompts for diffusion models. However, they are costly to train and lack adaptability when the backbone or reward function changes.

**Text embedding update**  Beyond prompt tuning, the text embedding $\mathbf{c}$ itself plays a central role in aligning images with textual intent. Most diffusion models use frozen text encoders and apply the same fixed embeddings across all timesteps, limiting their ability to capture the evolving semantics [6, 27, 59]. We hypothesize that dynamically adapting text embeddings, by transforming static $\mathbf{c}$ into time-dependent $\mathbf{c}_t$, can improve semantic alignment.

Recent works have begun to investigate this direction. EBCA [40] updates text embeddings at each cross-attention layer via an energy-based objective but lacks global semantic control. P2L [8] directly optimizes text embeddings for an inverse problem objective. Other works focus on special token tuning, such as Textual Inversion [14] and MinorityPrompt [55], which learn special token embeddings for personalized or minority instance generation, sometimes extended to time-dependent variants [55]. However, these methods are task-specific and limited to special tokens. In contrast, DATE provides a general, training-free framework that dynamically updates the entire text embedding throughout the diffusion sampling process, enabling fine-grained semantic adaptation without retraining.

## 2.3 Evaluation on text-to-image generation

Given the trained model for text-to-image generation, it is necessary to define an evaluation function that measures how well the generated outputs match the text-conditional inputs. We refer to this as the *text-conditioned evaluation function*, which can also serve as an objective function to guide generation. Common metrics like CLIP score [16] and ImageReward [57] are well-suited for this purpose and are easily applicable in the data space, as illustrated in Figure 2. Formally, this function can be expressed as $h(\mathbf{x}_0; y) \in \mathbb{R}$, and $h$ measures the alignment between the image $\mathbf{x}_0$ and the corresponding text input $y$. For simplicity, we assume that higher $h$ values indicate better

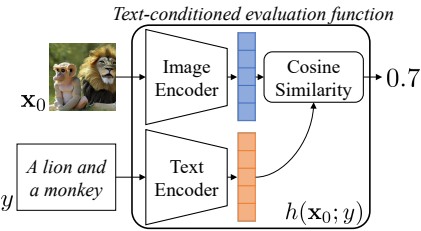

Figure 2: Examples of text-conditioned evaluation function.

alignment, implying stronger semantic consistency between images and texts.

Figure 1b plots the evaluation score changes over time for the mean predicted image, comparing 100 samples generated with fixed text embeddings and proposed dynamic text embeddings using Stable Diffusion. We observe two limitations of the previous approach: 1) the evaluation function $h$ is used only for evaluation without being incorporated into the sampling process, and 2) $h$ is only assessed at the final sample. To address these, we directly leverage $h$ as the learning objective during the intermediate periods of the sampling process. This update to the text embedding improves the evaluation function value, ultimately enhancing the quality of text-to-image generation.

# 3 Methods

## 3.1 Diffusion Adaptive Text Embedding (DATE)

In this section, we propose an objective function to optimize text embeddings during the sampling process. The diffusion sampling process can be expressed as follows:

$$\mathbf{x}_T \sim p_T(\mathbf{x}_T), \mathbf{x}_{t-1} \sim p_{\boldsymbol{\theta}}(\mathbf{x}_{t-1}|\mathbf{x}_t, \mathbf{c}_t) \text{ for } t = T, \cdots, 1, \tag{5}$$

where $p_{\boldsymbol{\theta}}(\mathbf{x}_{t-1}|\mathbf{x}_t, \mathbf{c}_t)$ is the distribution of the intermediate sample $\mathbf{x}_{t-1}$ at timestep $t-1$, generated from the sample $\mathbf{x}_t$ at timestep $t$ and the text embedding $\mathbf{c}_t$. Typically, the text embedding $\mathbf{c}_t$ is obtained from a pre-trained text encoder $\mathbf{I}_\phi$, i.e., $\mathbf{c}_{\text{org}} := \mathbf{I}_\phi(y)$, and remains fixed across all diffusion timesteps, i.e., $\mathbf{c}_t = \mathbf{c}_{\text{org}}$ for all $t$ (upper part of Figure 1a). However, we claim that the text embedding that is capable of producing effective generation output varies depending on the diffusion timestep $t$ and the current perturbed data $\mathbf{x}_t$ (lower part of Figure 1a). We refer to this dynamic text embedding as *Diffusion Adaptive Text Embedding (DATE)*.

Our goal is to find the text embedding $\mathbf{c}_t(\mathbf{x}_t)$[1] that maximizes the evaluation function $h$ for samples generated by the diffusion sampling process $p_{\boldsymbol{\theta}}$. The objective of DATE is expressed as follows:

$$\max_{\mathbf{c}_{1:T}} \mathbb{E}_{\mathbf{x}_T \sim p_T, \mathbf{x}_{0:T-1} \sim \prod_{\tau=1}^T p_{\boldsymbol{\theta}}(\mathbf{x}_{\tau-1}|\mathbf{x}_\tau, \mathbf{c}_\tau)}[h(\mathbf{x}_0; y)]. \tag{6}$$

It should be noted that Eq. (6) optimizes the alignment $h$ only at the final reverse diffusion step, so the other latent variables $\mathbf{x}_{1:T}$ are not directly regularized in any specific direction. However, the below derivation from Eqs. (7) to (10) shows that the optimization direction for each $\mathbf{x}_t$ can be derived from Eq. (6), which optimizes only $\mathbf{x}_0$, considering that $\mathbf{x}_0$ is the sequential sampling result from $\mathbf{x}_{1:T}$.

Since the sampling process of diffusion models proceeds iteratively from timestep $T$ to 0, we likewise aim to iteratively update the text embedding $\mathbf{c}_t$ according to this sequential order. Hence, this sequential sampling, which is required for practical implementation, renders Eq. (6) into Eq. (7). Specifically, the sequential nature of diffusion sampling requires two reformulations of Eq. (6).

**Motivation 1:** The adaptive text embedding $\mathbf{c}_t$ needs to be determined sequentially because the corresponding image is estimated at timestep $t$. Thus, the sequential decision on $\mathbf{c}_t$ decomposes the joint optimization into the step-wise optimization. Particularly, the diffusion sampling process requires a specific order in sequential decisions, i.e., from $T$ to 0. This motivation of decomposed and ordered optimization is reflected in the separated *max* operator in Eq. (7).

**Motivation 2:** The evaluation of $h$ is performed at the final sampling step 0, not at intermediate step $t$. For simplicity, we need to calculate the text-image alignment $h$ on the final generation result while maintaining $\mathbf{c}_t$. This motivates maintaining $\mathbf{c}_t$ until timestep 0 of the data distribution, which is reflected in the equality constraints on the feasible set $\mathcal{C}_t$ of each *max* operator in Eq. (7).

$$\max_{\mathbf{c}_1 \in \mathcal{C}_1} \cdots \max_{\mathbf{c}_t \in \mathcal{C}_t} \cdots \max_{\mathbf{c}_T \in \mathcal{C}_T} \mathbb{E}_{\mathbf{x}_T \sim p_T, \mathbf{x}_{0:T-1} \sim \prod_{\tau=1}^T p_{\boldsymbol{\theta}}(\mathbf{x}_{\tau-1}|\mathbf{x}_\tau, \mathbf{c}_\tau)}[h(\mathbf{x}_0; y)] \tag{7}$$

Here, $\mathcal{C}_t := \{\mathbf{c}_t : ||\mathbf{c}_t - \mathbf{c}_{\text{org}}||_2 \leq \rho, \mathbf{c}_\tau = \mathbf{c}_t \ \forall \tau < t\}$, $\mathbf{c}_{\text{org}}$ is the original text embedding, and $\rho$ is a scale hyperparameter. The constraint $||\mathbf{c}_t - \mathbf{c}_{\text{org}}||_2 \leq \rho$ keeps the optimized text embedding $\mathbf{c}_t$ does not deviate significantly from the original embedding $\mathbf{c}_{\text{org}}$, preserving its original semantic meaning.

Specifically, the optimization problem for $\mathbf{c}_t$ in Eq. (7) can be derived as follows:

$$\max_{\mathbf{c}_t \in \mathcal{C}_t} \mathbb{E}_{\mathbf{x}_{0:t-1} \sim \prod_{\tau=1}^t p_{\boldsymbol{\theta}}(\mathbf{x}_{\tau-1}|\mathbf{x}_\tau, \mathbf{c}_\tau)}[h(\mathbf{x}_0; y)] \Leftrightarrow \max_{\mathbf{c}_t : ||\mathbf{c}_t - \mathbf{c}_{\text{org}}||_2 \leq \rho} \mathbb{E}_{\mathbf{x}_0 \sim p_{\boldsymbol{\theta}}(\mathbf{x}_0|\mathbf{x}_t, \mathbf{c}_t)}[h(\mathbf{x}_0; y)]. \tag{8}$$

On the left-hand side of Eq. (8), the terms from timesteps from $t+1$ to $T$ are eliminated since $\mathbf{c}_{t+1:T}$ are set by the inner optimizations. On the right-hand side, the simplification arises from the constraint that all text embeddings from $t$ to 1 are identical. Therefore, at timestep $t$, our objective can be expressed as the expectation of the text-conditioned evaluation function with respect to $p_{\boldsymbol{\theta}}(\mathbf{x}_0|\mathbf{x}_t, \mathbf{c}_t)$.

However, solving Eq. (8) is computationally challenging. Evaluating the objective requires Monte Carlo sampling of $\mathbf{x}_0$ from $\mathbf{x}_t$, and each sample involves iterative sampling with multiple network evaluations, resulting in high computational cost. To address this, we apply a first-order Taylor approximation of the text-conditioned evaluation function $h$ around $\bar{\mathbf{x}}_0 := \bar{\mathbf{x}}_0(\mathbf{x}_t, \mathbf{c}_t; \boldsymbol{\theta}) = \mathbb{E}_{\mathbf{x}_0 \sim p_{\boldsymbol{\theta}}(\mathbf{x}_0|\mathbf{x}_t, \mathbf{c}_t)}[\mathbf{x}_0]$, a technique commonly used in previous studies [2, 8]:

$$\mathbb{E}_{\mathbf{x}_0}[h(\mathbf{x}_0; y)] \approx h(\bar{\mathbf{x}}_0; y) + \mathbb{E}_{\mathbf{x}_0}\left[(\mathbf{x}_0 - \bar{\mathbf{x}}_0)^T \nabla_{\mathbf{x}} h(\bar{\mathbf{x}}_0; y)\right] = h(\bar{\mathbf{x}}_0; y). \tag{9}$$

---

[1]For simplicity, we omit the dependency on $\mathbf{x}_t$ and write $\mathbf{c}_t(\mathbf{x}_t)$ as $\mathbf{c}_t$ whenever no ambiguity arises.

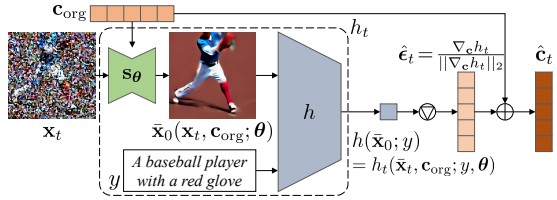

Figure 3: Update step of DATE at timestep $t$. The symbol with an inverted triangle inside a circle represents the normalized gradient with respect to $\mathbf{c}$, and $\oplus$ denotes summation.

**Algorithm 1** Diffusion Sampling with DATE

1: $\mathbf{x}_T \sim p_T(\cdot)$
2: $\mathbf{c}_{\text{org}} \leftarrow \mathbf{I}_\phi(y)$
3: **for** $t = T$ **to** $1$ **do**
4:      **if** $t \in \{\text{text update steps}\}$ **then**
5:          $\mathbf{c} \leftarrow \mathbf{c}_{\text{org}} + \rho \frac{\nabla_{\mathbf{c}} h_t(\mathbf{x}_t, \mathbf{c}_{\text{org}}; y, \boldsymbol{\theta})}{||\nabla_{\mathbf{c}} h_t(\mathbf{x}_t, \mathbf{c}_{\text{org}}; y, \boldsymbol{\theta})||_2}$
6:      **end if**
7:      $\mathbf{x}_{t-1} \leftarrow \mathbf{x}_t + \frac{1}{2}\beta_t(\mathbf{x}_t + \mathbf{s}_{\boldsymbol{\theta}}(\mathbf{x}_t, \mathbf{c}, t))$
8: **end for**

The equality in Eq. (9) holds because $\mathbb{E}_{\mathbf{x}_0}[\mathbf{x}_0 - \bar{\mathbf{x}}_0] = 0$. Therefore, through the approximation in Eq. (9), we propose the following alternative objective instead of Eq. (8):

$$\max_{\mathbf{c}_t:||\mathbf{c}_t - \mathbf{c}_{\text{org}}||_2 \leq \rho} h(\bar{\mathbf{x}}_0(\mathbf{x}_t, \mathbf{c}_t; \boldsymbol{\theta}); y) =: h_t(\mathbf{x}_t, \mathbf{c}_t; y, \boldsymbol{\theta}). \tag{10}$$

This objective in Eq. (10) optimizes the text-conditioned evaluation function on the mean predicted image $\bar{\mathbf{x}}_0$ given the current perturbed image $\mathbf{x}_t$ and the text embedding $\mathbf{c}_t$. Using the Tweedie's formula [10], the mean predicted image $\bar{\mathbf{x}}_0$ can be computed via a single score network evaluation:

$$\bar{\mathbf{x}}_0(\mathbf{x}_t, \mathbf{c}_t; \boldsymbol{\theta}) = (\mathbf{x}_t + (1 - \bar{\alpha}_t)\mathbf{s}_{\boldsymbol{\theta}}(\mathbf{x}_t, \mathbf{c}_t, t)) / \sqrt{\bar{\alpha}_t}, \tag{11}$$

where $\mathbf{s}_{\boldsymbol{\theta}}(\mathbf{x}_t, \mathbf{c}_t, t)$ is a conditional score network, and $\bar{\alpha}_t := \prod_{\tau=1}^t (1 - \beta_\tau)$ is the constant related to the variance schedule of the diffusion process.

## 3.2 Update process of DATE

We present an update process of text embeddings at each timestep to optimize the objective in Eq. (10) using the current perturbed data. Since evaluating this objective requires score network propagation (see Eq. 11), computational cost increases with each update. Therefore, inspired by sharpness-aware minimization [13], we estimate the updated text embedding $\mathbf{c}_t$ in a single update.

We decompose the updated embedding $\mathbf{c}_t$ into its original embedding $\mathbf{c}_{\text{org}}$ and an update direction $\boldsymbol{\epsilon}_t$:

$$\mathbf{c}_t = \mathbf{c}_{\text{org}} + \boldsymbol{\epsilon}_t. \tag{12}$$

By expressing $\mathbf{c}_t$ in terms of an update direction $\boldsymbol{\epsilon}_t$, we reformulate the optimization problem in Eq. (10) into an equivalent problem over $\boldsymbol{\epsilon}_t$, as shown in Eq. (13). Next, we approximate $h_t(\cdot, \mathbf{c}_{\text{org}} + \boldsymbol{\epsilon}_t; \cdot, \cdot)$ using a first-order Taylor expansion around $\boldsymbol{\epsilon}_t = \mathbf{0}$ (see the appendix for derivation):

$$\boldsymbol{\epsilon}_t^* := \arg\max_{||\boldsymbol{\epsilon}_t||_2 \leq \rho} h_t(\mathbf{x}_t, \mathbf{c}_{\text{org}} + \boldsymbol{\epsilon}_t; y, \boldsymbol{\theta}) \approx \arg\max_{||\boldsymbol{\epsilon}_t||_2 \leq \rho} \boldsymbol{\epsilon}_t^{\mathsf{T}} \nabla_{\mathbf{c}} h_t(\mathbf{x}_t, \mathbf{c}_{\text{org}}; y, \boldsymbol{\theta}) =: \hat{\boldsymbol{\epsilon}}_t. \tag{13}$$

The solution to this optimization problem can be derived using the Cauchy-Schwarz inequality. Consequently, the estimator for the optimal text embedding $\hat{\mathbf{c}}_t$ is given by

$$\hat{\boldsymbol{\epsilon}}_t = \rho \frac{\nabla_{\mathbf{c}} h_t(\mathbf{x}_t, \mathbf{c}_{\text{org}}; y, \boldsymbol{\theta})}{||\nabla_{\mathbf{c}} h_t(\mathbf{x}_t, \mathbf{c}_{\text{org}}; y, \boldsymbol{\theta})||_2}; \quad \hat{\mathbf{c}}_t = \mathbf{c}_{\text{org}} + \rho \frac{\nabla_{\mathbf{c}} h_t(\mathbf{x}_t, \mathbf{c}_{\text{org}}; y, \boldsymbol{\theta})}{||\nabla_{\mathbf{c}} h_t(\mathbf{x}_t, \mathbf{c}_{\text{org}}; y, \boldsymbol{\theta})||_2}. \tag{14}$$

This update step refines $\mathbf{c}_t$ by adjusting it along the normalized gradient direction of $h_t$, maximizing semantic alignment at the diffusion timestep $t$ under the current perturbed data $\mathbf{x}_t$. As a result, the updated text embedding $\hat{\mathbf{c}}_t$ dynamically adapts to the specific diffusion timestep and the corresponding perturbed data. Figure 3 visualizes this update step, and Algorithm 1 presents the overall algorithm for the diffusion sampling process with DATE. We introduce text embedding update steps in lines 4-6 of Algorithm 1, and the denoising process (line 7) can be performed using various diffusion samplers.

## 3.3 Theoretical analysis

We provide theoretical analyses of DATE, including performance guarantees and its influence on the data space. We provide proofs and additional approximation error analyses in Appendix A.

First, we show that both unconstrained and constrained optimizations of the text embedding in Eqs. (6) and (7) produce a better text embedding than the fixed text embedding $\mathbf{c}_{\text{org}}$.

**Proposition 1.** *Let* $h(\mathbf{c}_1, \cdots, \mathbf{c}_T) := \mathbb{E}_{\mathbf{x}_{0:T}}[h(\mathbf{x}_0; y)]$ *where* $\mathbf{x}_{0:T-1} \sim \prod_{\tau=1}^{T} p_{\boldsymbol{\theta}}(\mathbf{x}_{\tau-1}|\mathbf{x}_\tau, \mathbf{c}_\tau)$, $\mathbf{x}_T \sim p_T$, *and* $\mathcal{C}_t := \{\mathbf{c}_t : ||\mathbf{c}_t - \mathbf{c}_{\mathrm{org}}||_2 \leq \rho, \mathbf{c}_\tau = \mathbf{c}_t \, \forall \tau < t\}$. *Then,*

$$\max_{\mathbf{c}_{1:T}} h(\mathbf{c}_1, \cdots, \mathbf{c}_T) = \max_{\mathbf{c}_1} \cdots \max_{\mathbf{c}_t} \cdots \max_{\mathbf{c}_T} h(\mathbf{c}_1, \cdots, \mathbf{c}_T) \tag{15}$$

$$\geq \max_{\mathbf{c}_1 \in \mathcal{C}_1} \cdots \max_{\mathbf{c}_t \in \mathcal{C}_t} \cdots \max_{\mathbf{c}_T \in \mathcal{C}_T} h(\mathbf{c}_1, \cdots, \mathbf{c}_T) \geq h(\mathbf{c}_{\mathrm{org}}, \cdots, \mathbf{c}_{\mathrm{org}}). \tag{16}$$

The first equation shows that sequential maximization in Eq. (15) (corresponding to Motivation 1) attains the same optimum as the joint maximization. Introducing the constraints in Eq. (16) (corresponding to Motivation 2) restricts the feasible set and can lower the optimum. Nonetheless, Proposition 1 guarantees that both the unconstrained optimum (Eq. 6) and the constrained optimum (Eq. 7) yield a value at least as high as the fixed embedding. Because DATE is derived by approximating the optimization problem in Eq. (7), it is expected to improve the text-image alignment of the generated images compared to the fixed embedding.

Next, we present Theorem 2 to illustrate how the DATE update influences the perturbed data.

**Theorem 2.** *The score function for the text embedding* $\mathbf{c}_t$ *updated by DATE can be expressed as:*

$$\nabla_{\mathbf{x}_t} \log p_{\boldsymbol{\theta}}(\mathbf{x}_t | \hat{\mathbf{c}}_t) = \nabla_{\mathbf{x}_t} \log p_{\boldsymbol{\theta}}(\mathbf{x}_t | \mathbf{c}_{\mathrm{org}}) + \rho \nabla_{\mathbf{x}_t} \left\{ \frac{\nabla_{\mathbf{c}} h_t(\mathbf{x}_t, \mathbf{c}_{\mathrm{org}})^T}{||\nabla_{\mathbf{c}} h_t(\mathbf{x}_t, \mathbf{c}_{\mathrm{org}})||_2} \nabla_{\mathbf{c}} \log p_{\boldsymbol{\theta}}(\mathbf{x}_t | \mathbf{c}_{\mathrm{org}}) \right\} + O(\rho^2). \tag{17}$$

According to Theorem 2, the updated text embedding $\mathbf{c}_t$ can be interpreted as introducing a guidance term to the original score function, under a sufficiently small $\rho$. This guidance effectively improves the alignment between the evaluation function $h_t$ and the model likelihood from the perspective of the text embedding. Therefore, embedding-based guidance balances semantic alignment with the underlying model distribution, enhancing prompt fidelity without reducing generation quality.

### 3.4 Practical implementation

**Reducing computational cost** Updating the text embedding via Eq. (14) requires computing the gradient of $h_t$ through both diffusion and evaluation networks, which increases computational cost, as discussed in Appendix D.1. To mitigate this overhead, we update embeddings only at a subset of sampling steps (line 4 in Algorithm 1) and reuse the last update embeddings between updates, balancing performance and efficiency as shown in Figure 9 of Section 4.2. In addition, general computationally efficient strategies such as half-precision inference can be applied during sampling. As shown in Table 4, DATE remains compatible with such efficiency techniques, effectively reducing runtime and GPU memory consumption while incurring only a slight performance degradation.

**Selection of original text embedding** We explore two strategies for choosing the original text embedding $\mathbf{c}_{\mathrm{org}}$ at each update step: (1) always use the pre-trained text encoder output, $\mathbf{I}_\phi(y)$, preserving semantic integrity, and (2) use the embedding from the previous step, $\mathbf{c}_{t+1}$, allowing broader exploration of the embedding space. To prevent semantic drift in the second approach, we add an L2 regularizer $||\mathbf{c}_t - \mathbf{I}_\phi(y)||_2$ to the objective in Eq. (10). This term ensures update embeddings remain close to the original. Ablation results comparing these strategies also appear in Figure 9.

## 4 Experiments

We evaluate DATE for text-to-image generation primarily using U-Net-based Stable Diffusion (SD) v1.5 [47] with a pre-trained CLIP ViT-L/14 text encoder [42]. Additionally, to demonstrate broader applicability, we include evaluations on the latest transformer-based model, PixArt-$\alpha$ [5]. Unless stated otherwise, we set the text-conditioned evaluation function $h$ to CLIP score, the scale hyperparameter $\rho$ to 0.5, and use the embedding from the previous update as the original text embedding $\mathbf{c}_{\mathrm{org}}$ for each update step. Additional details are in Appendix C.

### 4.1 Quantitative results

Following previous evaluation protocols [38, 39, 58], we generate 5,000 images from randomly sampled captions in the COCO [30] validation set. We use DDIM [51] and DDPM [18] for SD, and DPM-Solver [32] for PixArt-$\alpha$. We evaluate the image quality and semantic alignment using several metrics, including zero-shot FID [17], CLIP score (CS) [16], and ImageReward (IR) [57], with detailed explanations provided in Appendix C.1.

Table 1: Performance on the COCO validation set. Sampling steps are indicated in parentheses, and *Time* is the average sampling time (min.) for 64 samples. **Bold** values indicate the best performance.

| Backbone | Method | Time | FID↓ | CLIP score↑ | ImageReward↑ |
|---|---|---|---|---|---|
| SD v1.5 [47] w/ DDIM [51] | Fixed text embedding (50 steps) | 5.64 | 18.66 | 0.3204 | 0.2132 |
| | Fixed text embedding (70 steps) | 7.87 | 18.27 | 0.3199 | 0.2137 |
| | EBCA [40] | 8.10 | 25.85 | 0.2877 | -0.3128 |
| | Universal Guidance [2] | 8.25 | 18.56 | 0.3216 | 0.2221 |
| | **DATE (50 steps)** | | | | |
| | 10% update with CLIP score | 7.82 | 17.90 | 0.3237 | 0.2364 |
| | all updates with CLIP score | 24.20 | **17.22** | **0.3292** | 0.2277 |
| | 10% update with ImageReward | 7.82 | 18.61 | 0.3224 | 0.4792 |
| | all updates with ImageReward | 24.20 | 18.17 | 0.3224 | **1.2972** |
| PixArt-$\alpha$ [5] w/ DPM-Solver [32] | Fixed text embedding (20 steps) | 4.35 | 31.07 | 0.3201 | 0.8140 |
| | Fixed text embedding (45 steps) | 9.03 | 30.62 | 0.3199 | 0.8174 |
| | **DATE (20 steps)** | | | | |
| | 50% update with CLIP score | 8.93 | **30.55** | **0.3237** | 0.8287 |
| | 50% update with ImageReward | 8.95 | 31.07 | 0.3221 | **0.9514** |

Table 2: Results on COCO using SD v1.5 with various evaluation functions. **Bold** values indicate the best performance, while *italic* values denote cases that underperform the fixed embedding.

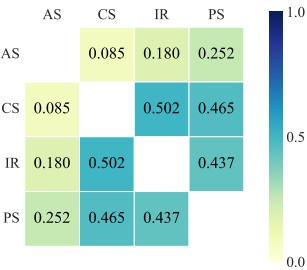

Figure 4: Pairwise Pearson correlations between evaluation functions.

| Method | Fidelity | | Semantic | Preference | |
|---|---|---|---|---|---|
| | FID↓ | AS↑ | CS↑ | IR↑ | PS↑ |
| Fixed embedding (50 steps) | 18.66 | 5.38 | 0.3204 | 0.2132 | 21.51 |
| Fixed embedding (70 steps) | 18.27 | 5.37 | 0.3199 | 0.2137 | 21.50 |
| **DATE (50 steps , 10% update)** | | | | | |
| with Aesthetic Score (AS) | *18.82* | **5.58** | *0.3169* | *0.1910* | *21.46* |
| with CLIP Score (CS) | **17.90** | *5.35* | **0.3237** | 0.2364 | 21.53 |
| with ImageReward (IR) | 18.61 | 5.40 | 0.3224 | **0.4792** | 21.53 |
| with PickScore (PS) | 18.49 | 5.42 | 0.3225 | 0.2745 | **21.93** |

**Main results** Table 1 compares the performance of different text embeddings across various backbones and samplers. Tables 8 and 9 in Appendix D.1 further extend this comparison to different classifier-free guidance scales and sampling methods. Across all settings, DATE consistently outperforms the fixed text embedding, even with matched sampling time for a fair comparison. Notably, DATE improves the metric used for its evaluation function $h$, as well as other metrics, suggesting that it enhances the overall text-conditional generation quality beyond optimizing a single objective.

Compared to recent methods, EBCA [40] applies the energy-based optimization to text embeddings within cross-attention layers but yields suboptimal performance. Universal Guidance [2], which injects $h_t$-based guidance directly into the data space, yields performance gains but still falls short of DATE. In contrast, DATE explicitly updates text embeddings to optimize the evaluation objective, achieving better semantic alignment (CS, IR) while preserving generative quality (FID), as supported by our theoretical analysis.

**Evaluation function** To better understand the role of evaluation functions, we analyze DATE under different functions. Specifically, we examine pairwise correlations among different functions and evaluate DATE when each metric, individually or in combination, serves as the evaluation function. We consider Aesthetic Score (AS) [49] for image fidelity, CS for semantic alignment, and two human-preference-based metrics, IR and PickScore (PS) [26]. Each provides per-instance scores for generated images, enabling both correlation analysis and direct integration with DATE.[2]

Figure 4 presents Pearson correlations computed from 1,000 Stable Diffusion samples, and Figure 5a visualizes instance-level relationships between CS and IR. AS and CS exhibit minimal correlation,

---

[2]FID is excluded since it measures distribution-level similarity and lacks instance-level scores.

Table 3: Results on COCO across backbones.

| Backbone | Methods | FID↓ | CS↑ | IR↑ |
|---|---|---|---|---|
| SD3 [11] | Fixed | **26.00** | 0.3337 | 1.0018 |
| | **DATE (ours)** | **26.00** | **0.3340** | **1.0457** |
| FLUX [28] | Fixed | 29.59 | 0.3257 | 0.9634 |
| | **DATE (ours)** | **29.41** | **0.3283** | **0.9768** |
| SDXL [41] | Fixed | 18.27 | 0.3368 | 0.7284 |
| | **DATE (ours)** | **18.03** | **0.3382** | **0.9096** |

Table 4: Results on COCO with half-precision.

| Method | Time | Memory | FID↓ | CS↑ | IR↑ |
|---|---|---|---|---|---|
| Fixed embedding (50 steps) | 5.64 | 24.0 | 18.66 | 0.3204 | 0.2132 |
| Fixed embedding (70 steps) | 7.87 | 24.0 | 18.27 | 0.3199 | 0.2137 |
| **DATE (50 steps , 10% update)** | | | | | |
| with CLIP Score (CS) | 7.82 | 61.5 | 17.90 | 0.3237 | 0.2364 |
| with CLIP Score (CS) (FP16) | 4.40 | 32.9 | 17.99 | 0.3229 | 0.2265 |
| with ImageReward (IR) | 7.82 | 61.5 | 18.61 | 0.3224 | 0.4792 |
| with ImageReward (IR) (FP16) | 4.02 | 30.6 | 18.03 | 0.3222 | 0.4773 |

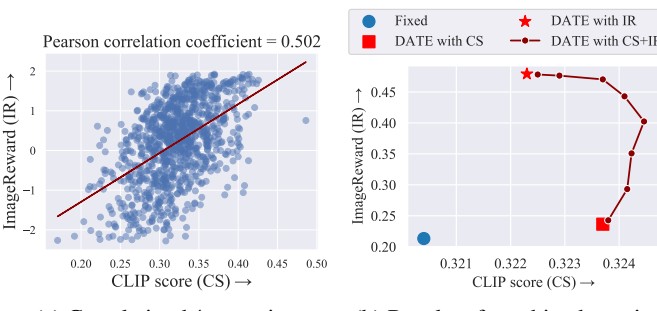

(a) Correlation b/w metrics    (b) Results of combined metrics

Figure 5: (a) Correlation between CLIP score (CS) and ImageReward (IR). Blue dots represent individual samples, and the red line indicates their linear regression line. (b) Trade-off between CS and IR. For *DATE with CS+IR*, we use a weighted sum of CS and IR as the evaluation function $h$, and we plot the performance changes as the weights are varied.

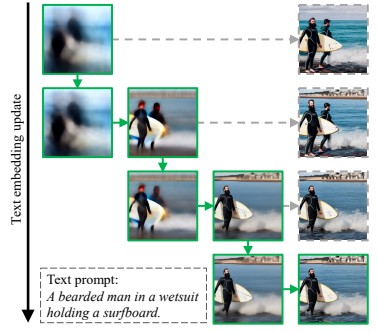

Figure 6: Mean predicted images $\bar{\mathbf{x}}_0(\mathbf{x}_t, \mathbf{c}_t)$ for each diffusion timestep and text embedding during the sampling process with DATE.

suggesting independence between aesthetic quality and semantic alignment. IR and PS correlate moderately with CS and weakly with AS, indicating that they capture both fidelity and alignment aspects. IR and PS themselves correlate moderately, reflecting their distinct sensitivities.

With these metrics as the evaluation function, DATE generally improves all metrics over the fixed embedding, as shown in Table 2. However, when AS is used, other metrics often degrade, consistent with its low semantic correlation. Moreover, combining multiple evaluation functions, such as a weighted sum of CS and IR, improves both metrics simultaneously, as shown in Figure 5b and Table 10. DATE thus remains compatible with multi-metric objectives, where adjusting weights balances trade-off. Interestingly, the combined objective can yield even higher CS than for CS alone, demonstrating the synergistic potential among evaluation functions during test-time optimization.

**Applicability to other backbones** DATE can be applied on top of powerful base models. We evaluate it on recent architectures, including SD3 [11], FLUX [28], and SDXL [41], following the default configurations described in Appendix C.1. As shown in Table 3, DATE consistently improves text-image alignment and generation quality across all these models, demonstrating its robustness and broad applicability to modern diffusion architectures.

**Computational efficiency** To mitigate the increased computational cost introduced by gradient computations, we explore memory-efficient sampling strategies. In particular, we apply half-precision inference during sampling, which substantially reduces runtime and memory consumption while maintaining competitive performance, as shown in Table 4. Notably, casting the CLIP model used in the evaluation function to half-precision led to performance degradation, so we retain it in full precision; however, since the diffusion model dominates computational cost, applying half-precision to remaining components still provides significant savings. These results demonstrate that DATE remains fully compatible with standard memory-efficient strategies.

## 4.2 Analysis of DATE

**Generation process** The green boxes in Figure 6 show the generation process of DATE. Fixed text embeddings misinterpret '*a man*' as '*two men*', but DATE corrects this by dynamic updates.

Table 5: Ablation studies.

| Method | FID↓ | CS↑ | IR↑ |
|---|---|---|---|
| Fixed embedding | 18.66 | 0.3204 | 0.2132 |
| Random update | 18.66 | 0.3204 | 0.2136 |
| $h(\mathbf{x}_t; y)$ | 18.80 | 0.3200 | 0.2121 |
| Unnormalized gradient | 18.46 | 0.3212 | 0.2225 |
| **DATE (ours)** | **17.91** | **0.3220** | **0.2229** |

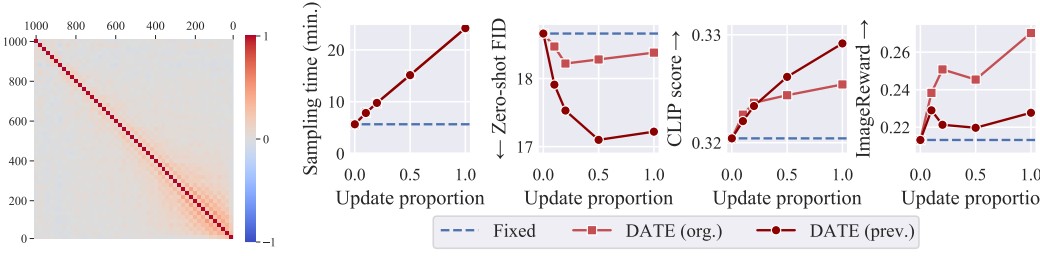

Figure 7: Sensitivity analysis on the scale $\rho$.

Figure 8: Cosine similarity of update directions $\hat{\epsilon}_t$ between timesteps.

Figure 9: Comparison of sampling time and performance based on the proportion of update steps to total sampling steps.

**Time- and instance-adaptive text embedding**  We analyze the time-dependent text embeddings by measuring the cosine similarity between update directions $\hat{\epsilon}_t$ at different timesteps, averaged over 100 samples, in Figure 8. Most timestep pairs show near-zero similarity, with about 85% of pairs below 0.1, indicating that optimal embeddings differ across timesteps. In contrast, adjacent timesteps generally show positive similarity, suggesting that reusing the embedding from the previous step can reduce runtime with a moderate loss in performance.

We also examine instance-specific adaptation by measuring the cosine similarity of update directions across different samples of the same prompt at each timestep. The similarity remains close to zero (below 0.05) across all timesteps, showing that each instance has distinct text embedding updates, reinforcing the need for adaptive embeddings in text-to-image generation.

**Ablation studies**  Table 5 presents several ablations for the text embedding update. *Random update*, which replaces the gradient with a random Gaussian vector, performs similarly to the fixed embedding. This indicates that our update is not just a perturbation, but plays a meaningful role in text-image alignment. Alignment with perturbed data, which aligns the text prompt with the perturbed data $\mathbf{x}_t$ using $h(\mathbf{x}_t; y)$, results in degraded performance, likely because the evaluation function $h$ lacks explicit information in the perturbed data space. *Unnormalized gradient* performs better than fixed embeddings but remains inferior to DATE. This suggests that using the unnormalized gradient still serves as a gradient ascent method to improve $h_t$, but a single-step update is suboptimal.

**Sensitivity analysis of $\rho$**  Figure 7 presents a sensitivity analysis of the scale hyperparameter $\rho$, which controls the magnitude of $\hat{\epsilon}_t$. The performance consistently outperforms the fixed embedding, but higher $\rho$ causes degradation in some regions. This is likely due to errors in the Taylor approximation from an expanded feasible region in Eq. (13). Based on this, we set $\rho$ to 0.5 in our experiments.

**Selection of original embedding**  Figure 9 analyzes the effect of selecting the original embedding at each update step, discussed in Section 3.4. Initializing with the previously updated embedding generally improves the CLIP score, likely due to a broader exploration of the embedding space. Based on this, we adopt this strategy in our experiments.

**Embedding update steps**  Figure 9 also shows the sampling time and performance based on the number of update steps. We observe that even a few updates improve performance. Increasing the update proportion tends to further improve performance, but it also increases the sampling time.

Figure 10 compares different update strategies while keeping the total number of updates the same, using ImageReward as the objective. Here, $\mathbf{c}^{\mathrm{org}}$ is the fixed embedding; $\mathbf{c}^{\mathrm{u}}$ refers uniform updates; and $\mathbf{c}^{\mathrm{e}}$, $\mathbf{c}^{\mathrm{m}}$, and $\mathbf{c}^{\mathrm{l}}$ correspond to updates at early, middle, and late sampling steps, respectively. We find that updating at any sets improves performance over no update, with mid-to-late updates ($\mathbf{c}^{\mathrm{m}}$ and $\mathbf{c}^{\mathrm{l}}$) being more effective for text-image alignment. This suggests that adjusting text embeddings during the fine-grained detail refinement phase in the later sampling steps is more effective.

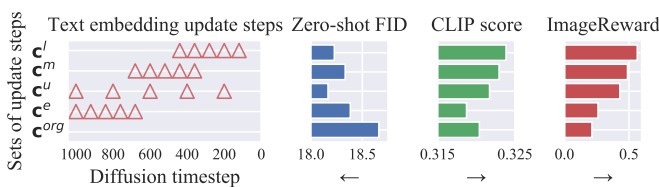

Figure 10: Comparison of update steps. On the left, the red triangles mark update steps, while the bars in the remaining graphs show the performance for each set of update steps.

Figure 11: Average similarity on the AnE dataset [4] for multi-concept generation.

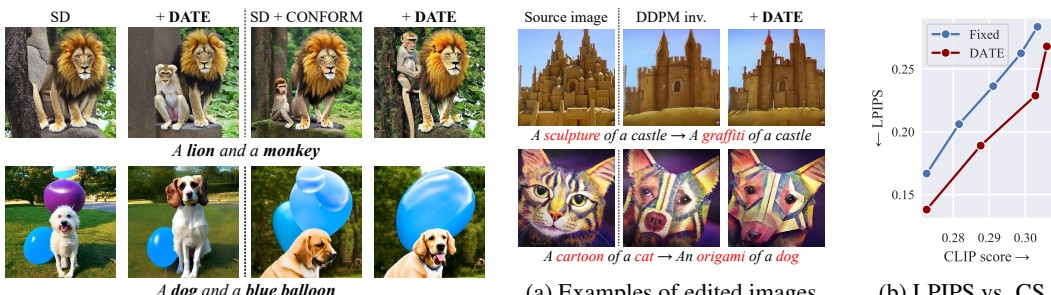

Figure 12: Examples of generated images on the AnE dataset for multi-concept generation.

(a) Examples of edited images
(b) LPIPS vs. CS

Figure 13: Fixed embedding versus DATE in DDPM inversion [21] for text-guided image editing.

### 4.3 Applications

**Multi-concept generation**    Multi-concept generation aims to generate multiple concepts (e.g., objects and attributes) within a single text prompt. We evaluate DATE on the AnE dataset [4] with 1) base SD and 2) SD with CONFORM [35], a method for multi-concept generation. Following previous work [4, 35], we generate 64 images per prompt and evaluate similarity between text and images using various metrics. Additional details are provided in Appendix C.2.

Figure 11 compares performance with and without DATE and shows that applying DATE consistently improves performance across all metrics. CONFORM provides better performance than DATE alone, but it requires explicit annotation of attribute-object pairs, and applying DATE to CONFORM can further improve performance. Figure 12 illustrates that DATE improves object representation under the same prompt and random seed. These results highlight the effectiveness of DATE in refining concept representation and improving text-image alignment. Additional results are in Appendix D.3.

**Text-guided image editing**    Text-guided image editing modifies an input image based on text prompts, allowing natural language adjustments [34, 37]. We apply DATE to DDPM inversion [21], a diffusion-based image editing model; and evaluate it on 30 source images from ImageNet-R-TI2I [54], each modified with five target prompts. Evaluation is based on LPIPS [60] for perceptual similarity with the source image and CLIP score for text-image alignment with the target prompt.

Figure 13b compares LPIPS and CLIP score across different guidance scales. The result shows that DATE achieves a better trade-off between fidelity to the source image and alignment with the target text. Figure 13a presents edited images for each method. These results suggest that DATE improves text-guided image editing by better balancing content preservation and textual modifications.

## 5    Conclusion

We propose Diffusion Adaptive Text Embedding (DATE), which improves text-to-image diffusion models by dynamically refining text embeddings throughout the diffusion sampling process. Unlike conventional methods with fixed embeddings from a frozen text encoder, DATE adaptively updates text representations at intermediate steps, effectively addressing semantic misinterpretations and improving text-image alignment. Experiments show that DATE consistently outperforms fixed embeddings across various tasks and methods involving text-to-image diffusion models, highlighting the potential of time- and instance-dependent text embeddings to improve text-to-image generation.

## Acknowledgments and Disclosure of Funding

This work was supported by the IITP (Institute of Information & Communications Technology Planning & Evaluation)-ITRC (Information Technology Research Center) grant funded by the Korea government (Ministry of Science and ICT) (IITP-2025-RS-2024-00437268).

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

# A Proof and additional theoretical analysis

## A.1 Proof of Proposition 1

**Proposition 1.** *Let* $h(\mathbf{c}_1, \cdots, \mathbf{c}_T) := \mathbb{E}_{\mathbf{x}_{0:T}}[h(\mathbf{x}_0; y)]$ *where* $\mathbf{x}_{0:T-1} \sim \prod_{\tau=1}^{T} p_{\boldsymbol{\theta}}(\mathbf{x}_{\tau-1}|\mathbf{x}_{\tau}, \mathbf{c}_{\tau})$, $\mathbf{x}_T \sim p_T$, *and* $\mathcal{C}_t := \{\mathbf{c}_t : ||\mathbf{c}_t - \mathbf{c}_{\mathrm{org}}||_2 \leq \rho, \mathbf{c}_{\tau} = \mathbf{c}_t \ \forall \tau < t\}$. *Then,*

$$\max_{\mathbf{c}_{1:T}} h(\mathbf{c}_1, \cdots, \mathbf{c}_T) = \max_{\mathbf{c}_1} \cdots \max_{\mathbf{c}_t} \cdots \max_{\mathbf{c}_T} h(\mathbf{c}_1, \cdots, \mathbf{c}_T) \tag{15}$$

$$\geq \max_{\mathbf{c}_1 \in \mathcal{C}_1} \cdots \max_{\mathbf{c}_t \in \mathcal{C}_t} \cdots \max_{\mathbf{c}_T \in \mathcal{C}_T} h(\mathbf{c}_1, \cdots, \mathbf{c}_T) \geq h(\mathbf{c}_{\mathrm{org}}, \cdots, \mathbf{c}_{\mathrm{org}}). \tag{16}$$

*Proof.* The first equality of Eq. (15) holds because the order of the maximization problems in the LHS of Eq. (15) can be interchanged. The second inequality, from the RHS of Eq. (15) to the LHS of Eq. (16), follows since both problems have the same objective function, but the LHS of Eq. (16) has a more restrictive feasible set. Finally, the last inequality from Eq. (16) holds because $(\mathbf{c}_{\mathrm{org}}, \cdots, \mathbf{c}_{\mathrm{org}})$ belongs to the feasible set of the LHS of Eq. (16), which ensures that the optimal value of the LHS of Eq. (16) is equal to or greater than $h(\mathbf{c}_{\mathrm{org}}, \cdots, \mathbf{c}_{\mathrm{org}})$. $\square$

## A.2 Proof of Theorem 2

**Theorem 2.** *The score function for the text embedding* $\mathbf{c}_t$ *updated by DATE can be expressed as:*

$$\nabla_{\mathbf{x}_t} \log p_{\boldsymbol{\theta}}(\mathbf{x}_t|\hat{\mathbf{c}}_t) = \nabla_{\mathbf{x}_t} \log p_{\boldsymbol{\theta}}(\mathbf{x}_t|\mathbf{c}_{\mathrm{org}}) + \rho \nabla_{\mathbf{x}_t} \left\{ \frac{\nabla_{\boldsymbol{c}} h_t(\mathbf{x}_t, \mathbf{c}_{\mathrm{org}})^T}{||\nabla_{\boldsymbol{c}} h_t(\mathbf{x}_t, \mathbf{c}_{\mathrm{org}})||_2} \nabla_{\mathbf{c}} \log p_{\boldsymbol{\theta}}(\mathbf{x}_t|\mathbf{c}_{\mathrm{org}}) \right\} + O(\rho^2). \tag{17}$$

*Proof.* By applying a first-order Taylor expansion, we obtain the following:

$$\log p_{\boldsymbol{\theta}}(\mathbf{x}_t|\hat{\mathbf{c}}_t) = \log p_{\boldsymbol{\theta}}\left(\mathbf{x}_t \Big| \mathbf{c}_{\mathrm{org}} + \rho \frac{\nabla_{\boldsymbol{c}} h_t(\mathbf{x}_t, \mathbf{c}_{\mathrm{org}})}{||\nabla_{\boldsymbol{c}} h_t(\mathbf{x}_t, \mathbf{c}_{\mathrm{org}})||_2}\right) \tag{18}$$

$$= \log p_{\boldsymbol{\theta}}(\mathbf{x}_t|\mathbf{c}_{\mathrm{org}}) + \rho \frac{\nabla_{\boldsymbol{c}} h_t(\mathbf{x}_t, \mathbf{c}_{\mathrm{org}})^T}{||\nabla_{\boldsymbol{c}} h_t(\mathbf{x}_t, \mathbf{c}_{\mathrm{org}})||_2} \nabla_{\mathbf{c}} \log p_{\boldsymbol{\theta}}(\mathbf{x}_t|\mathbf{c}_{\mathrm{org}}) + O(\rho^2). \tag{19}$$

Taking the gradient with respect to $\mathbf{x}_t$ on both side then confirms the statement:

$$\nabla_{\mathbf{x}_t} \log p_{\boldsymbol{\theta}}(\mathbf{x}_t|\hat{\mathbf{c}}_t) \tag{20}$$

$$= \nabla_{\mathbf{x}_t} \log p_{\boldsymbol{\theta}}\left(\mathbf{x}_t \Big| \mathbf{c}_{\mathrm{org}} + \rho \frac{\nabla_{\boldsymbol{c}} h_t(\mathbf{x}_t, \mathbf{c}_{\mathrm{org}})}{||\nabla_{\boldsymbol{c}} h_t(\mathbf{x}_t, \mathbf{c}_{\mathrm{org}})||_2}\right) \tag{21}$$

$$= \nabla_{\mathbf{x}_t} \log p_{\boldsymbol{\theta}}(\mathbf{x}_t|\mathbf{c}_{\mathrm{org}}) + \nabla_{\mathbf{x}_t} \left\{ \rho \frac{\nabla_{\boldsymbol{c}} h_t(\mathbf{x}_t, \mathbf{c}_{\mathrm{org}})^T}{||\nabla_{\boldsymbol{c}} h_t(\mathbf{x}_t, \mathbf{c}_{\mathrm{org}})||_2} \nabla_{\mathbf{c}} \log p_{\boldsymbol{\theta}}(\mathbf{x}_t|\mathbf{c}_{\mathrm{org}}) \right\} + O(\rho^2). \tag{22}$$

$\square$

## A.3 Detailed derivation of Eq. (13)

By applying a first-order Taylor approximation of $h_t$ around $\epsilon_t = 0$ in Eq. (23), we derive the expansion shown in Eq. (24). Since the first term in Eq. (24) is independent of $\epsilon_t$, it can be omitted from the optimization objective, as shown in Eq. (25).

$$\boldsymbol{\epsilon}_t^* := \underset{||\boldsymbol{\epsilon}_t||_2 \leq \rho}{\arg\max} \ h_t(\mathbf{x}_t, \mathbf{c}_{\mathrm{org}} + \boldsymbol{\epsilon}_t; y, \boldsymbol{\theta}) \tag{23}$$

$$\approx \underset{||\boldsymbol{\epsilon}_t||_2 \leq \rho}{\arg\max} \left\{ h_t(\mathbf{x}_t, \mathbf{c}_{\mathrm{org}}; y, \boldsymbol{\theta}) + \boldsymbol{\epsilon}_t^{\mathrm{T}} \nabla_{\boldsymbol{c}} h_t(\mathbf{x}_t, \mathbf{c}_{\mathrm{org}}; y, \boldsymbol{\theta}) \right\} \tag{24}$$

$$= \underset{||\boldsymbol{\epsilon}_t||_2 \leq \rho}{\arg\max} \ \boldsymbol{\epsilon}_t^{\mathrm{T}} \nabla_{\boldsymbol{c}} h_t(\mathbf{x}_t, \mathbf{c}_{\mathrm{org}}; y, \boldsymbol{\theta}) =: \hat{\boldsymbol{\epsilon}}_t. \tag{25}$$

### A.4 Theoretical analysis of approximation errors

When the text-conditioned evaluation function $h$ is the CLIP score, we analyze the approximation error for Eq. (9) analogously to the theoretical analysis presented in [8].

**Proposition 3.** *Let $h(\mathbf{x}_0; y) = g(\mathbf{f}_I(\mathbf{x}_0); \mathbf{f}_T(y))$ is the CLIP score, where $\mathbf{f}_I$ and $\mathbf{f}_T$ are CLIP image and text encoder, respectively, and $g$ is the cosine similarity. Assume that there exists a constant $K > 0$ such that $||\mathbf{f}_I(\mathbf{x}_0)|| \geq K$ for all $\mathbf{x}_0 \in \mathcal{X}_0$, where $\mathcal{X}_0$ is the support of $p_{\boldsymbol{\theta}}(\mathbf{x}_0|\mathbf{x}_t, \mathbf{c}_t)$. Then, the approximation error of Eq. (10) is upper bounded by:*

$$|\mathbb{E}_{\mathbf{x}_0 \sim p_{\boldsymbol{\theta}}(\mathbf{x}_0|\mathbf{x}_t, \mathbf{c}_t)}[h(\mathbf{x}_0; y)] - h(\bar{\mathbf{x}}_0; y)| \leq \frac{1}{K} \cdot \max_{\mathbf{x}_0 \in \mathcal{X}_0} ||\nabla_{\mathbf{x}} \mathbf{f}_I(\mathbf{x})|| \cdot m_1, \qquad (26)$$

*where $m_1 := \int ||\mathbf{x}_0 - \bar{\mathbf{x}}_0||p(\mathbf{x}_0|\mathbf{x}_t, \mathbf{c}_t)d\mathbf{x}_0$ is the mean deviation of the conditional distribution of $\mathbf{x}_0$.*

*Proof.* First, we prove the following lemma for the property of the cosine similarity.

**Lemma 4.** *Assume that there exists a constant $K > 0$ such that $||\mathbf{x}|| \geq K$ for all $\mathbf{x}$. Then,*

$$|g(\mathbf{x}; \mathbf{y}) - g(\mathbf{x}'; \mathbf{y})| \leq \frac{1}{K}||\mathbf{x} - \mathbf{x}'||. \qquad (27)$$

*Proof of Lemma.* The gradient of $g$ with respect to $\mathbf{x}$ is:

$$\nabla_{\mathbf{x}} g(\mathbf{x}; \mathbf{y}) = \frac{1}{||\mathbf{x}||||\mathbf{y}||}\left(\mathbf{y} - \frac{(\mathbf{y}^T\mathbf{x})}{||\mathbf{x}||^2}\mathbf{x}\right). \qquad (28)$$

Therefore, the norm of gradient can be derived as follows:

$$||\nabla_{\mathbf{x}} g(\mathbf{x}; \mathbf{y})|| = \frac{1}{||\mathbf{x}||||\mathbf{y}||}\left|\left|\mathbf{y} - \frac{(\mathbf{y}^T\mathbf{x})}{||\mathbf{x}||^2}\mathbf{x}\right|\right| \leq \frac{1}{||\mathbf{x}||||\mathbf{y}||}||\mathbf{y}|| = \frac{1}{||\mathbf{x}||} \leq \frac{1}{K}. \qquad (29)$$

Therefore,

$$|g(\mathbf{x}; \mathbf{y}) - g(\mathbf{x}'; \mathbf{y})| \leq (\max_{\mathbf{x}} ||\nabla_{\mathbf{x}} g(\mathbf{x}; \mathbf{y})||) \cdot ||\mathbf{x} - \mathbf{x}'|| = \frac{1}{K}||\mathbf{x} - \mathbf{x}'|| \qquad (30)$$

Note that the first inequality comes from the mean value inequality.

$\square$

From Lemma 4, we can derive the approximation error as follows.

$$|\mathbb{E}_{\mathbf{x}_0 \sim p_{\boldsymbol{\theta}}(\mathbf{x}_0|\mathbf{x}_t, \mathbf{c}_t)}[h(\mathbf{x}_0; y)] - h(\bar{\mathbf{x}}_0; y)| \leq \int |h(\mathbf{x}_0; y) - h(\bar{\mathbf{x}}_0; y)|p_{\boldsymbol{\theta}}(\mathbf{x}_0|\mathbf{x}_t, \mathbf{c}_t)d\mathbf{x}_0 \qquad (31)$$

$$= \int |g(\mathbf{f}_I(\mathbf{x}_0); \mathbf{f}_T(y)) - g(\mathbf{f}_I(\bar{\mathbf{x}}_0); \mathbf{f}_T(y))|p_{\boldsymbol{\theta}}(\mathbf{x}_0|\mathbf{x}_t, \mathbf{c}_t)d\mathbf{x}_0 \qquad (32)$$

$$\leq \frac{1}{K}\int ||\mathbf{f}_I(\mathbf{x}_0) - \mathbf{f}_I(\bar{\mathbf{x}}_0)||p_{\boldsymbol{\theta}}(\mathbf{x}_0|\mathbf{x}_t, \mathbf{c}_t)d\mathbf{x}_0 \qquad (33)$$

$$\leq \frac{1}{K} \cdot \max_{\mathbf{x}_0 \in \mathcal{X}_0} ||\nabla_{\mathbf{x}} \mathbf{f}_I(\mathbf{x})|| \cdot \int ||\mathbf{x}_0 - \bar{\mathbf{x}}_0||p_{\boldsymbol{\theta}}(\mathbf{x}_0|\mathbf{x}_t, \mathbf{c}_t)d\mathbf{x}_0 \qquad (34)$$

$$= \frac{1}{K} \cdot \max_{\mathbf{x}_0 \in \mathcal{X}_0} ||\nabla_{\mathbf{x}} \mathbf{f}_I(\mathbf{x})|| \cdot m_1 \qquad (35)$$

$\square$

In the upper bound of the approximation error in Eq. (26), $K$ is the minimum norm of CLIP image encoder outputs, which is about 25 in our experiments. Also, $\max ||\nabla_{\mathbf{x}} \mathbf{f}_I(\mathbf{x})||$ reflects the sharpness of CLIP image encoder; since the encoder is composed of neural networks, this value is finite, and

smoother image encoders result in a lower approximation error. $m_1$ decreases as $t$ becomes smaller. Therefore, as $t$ approaches 0, the approximation error is reduced.

Additionally, by Taylor's theorem, the approximation error for text embedding updates in Eq. (13) is

$$R_1(\boldsymbol{\epsilon}_t) := \frac{1}{2}\boldsymbol{\epsilon}_t^T H_{h_t}(\tilde{\mathbf{c}})\boldsymbol{\epsilon}_t = O(\rho^2), \tag{36}$$

where $H_{h_t}(\tilde{\mathbf{c}})$ is the Hessian of $h_t$ evaluated at some $\tilde{\mathbf{c}}$ between $\mathbf{c}_{\mathrm{org}}$ and $\mathbf{c}_{\mathrm{org}} + \boldsymbol{\epsilon}_t$. Since $||\boldsymbol{\epsilon}_t|| \leq \rho$, this error is $O(\rho^2)$. Therefore, tuning $\rho$ trades off approximation accuracy against optimization flexibility. We empirically analyze this trade-off in Figure 7.

### A.5 Theoretical analysis with convex text-conditioned evaluation function

In Section 3.1 of the main manuscript, we reformulate the optimization problem in Eq. (8) into Eq. (10) using the linear approximation in Eq. (9). At this point, when the text-conditioned evaluation function $h$ is convex, the objective functions of the two optimization problems satisfy the following inequality.

**Proposition 5.** *If $h$ is convex with respect to $\mathbf{x}_0$, then,*

$$h_t(\mathbf{x}_t, \mathbf{c}_t; y, \theta) := h(\bar{\mathbf{x}}_0(\mathbf{x}_t, \mathbf{c}_t; \theta); y) \leq \mathbb{E}_{\mathbf{x}_0 \sim p_{\boldsymbol{\theta}}(\mathbf{x}_0|\mathbf{x}_t, \mathbf{c}_t)}[h(\mathbf{x}_0; y)], \tag{37}$$

*where* $\bar{\mathbf{x}}_0(\mathbf{x}_t, \mathbf{c}_t; \boldsymbol{\theta}) := \mathbb{E}_{\mathbf{x}_0 \sim p_{\boldsymbol{\theta}}(\mathbf{x}_0|\mathbf{x}_t, \mathbf{c}_t)}[\mathbf{x}_0].$

*Proof.* This follows directly from Jensen's inequality for the convex function $h$. $\square$

Consequently, optimizing the text embedding $\mathbf{c}_t$ to maximize the proposed objective in Eq. (10) is expected to increase the target value $\mathbb{E}_{\mathbf{x}_0 \sim p_{\boldsymbol{\theta}}(\mathbf{x}_0|\mathbf{x}_t, \mathbf{c}_t)}[h(\mathbf{x}_0; y)]$ as well. However, Proposition 5 relies on the assumption of the convexity of $h$ with respect to $\mathbf{x}_0$. This assumption may not always hold in practice, since $h$ often contains a pre-trained neural network that introduces non-convexity.

## B Related works

### B.1 Diffusion models

We additionally provide an explanation of the stochastic differential equation (SDE) formulation of diffusion models. In continuous time space, the diffusion process can be generalized by formulating the forward and reverse processes as SDEs [53]. The forward process is formulated as:

$$\mathrm{d}\mathbf{x}_t = \mathbf{f}(\mathbf{x}_t, t)\mathrm{d}t + g(t)\mathrm{d}\mathbf{w}_t, \tag{38}$$

where $\mathbf{f}$ and $g$ are the drift and volatility functions, respectively. Here, $\mathbf{w}_t$ denotes the standard Wiener process and $t \in [0, T]$. Based on Eq. (38), a data instance $\mathbf{x}_0$ is gradually perturbed towards $\mathbf{x}_T$.

Once $\mathbf{f}$ and $g$ of the forward process are specified, the reverse process is uniquely defined as shown in Eq. (39), following the previous work [1]:

$$\mathrm{d}\mathbf{x}_t = [\mathbf{f}(\mathbf{x}_t, t) - g^2(t)\nabla_{\mathbf{x}_t} \log q_t(\mathbf{x}_t)]\mathrm{d}\bar{t} + g(t)\mathrm{d}\bar{\mathbf{w}}_t, \tag{39}$$

where $q_t(\mathbf{x}_t)$ is the marginal probability distribution of $\mathbf{x}_t$ at timestep $t$, and $\bar{\mathbf{w}}_t$ denotes the reverse-time Wiener process.

To generate samples, the reverse process requires an intermediate score function $\nabla_{\mathbf{x}_t} \log q_t(\mathbf{x}_t)$, which is generally intractable. Instead, a neural network $\mathbf{s}_\theta(\mathbf{x}_t, t)$ is used to approximate the score function via score matching loss [53]. Note that the score matching objective serves as an upper bound on the negative log-likelihood under certain temporal weighting functions [52]. This score matching objective is equivalent to the noise prediction [18, 9] or the data reconstruction [25, 22] objectives.

Table 6: Comparison of the diffusion guidance methods.

| Method | Guidance target | Guidance module |
|---|---|---|
| Classifier-free guidance [19] | Perturbed data $\mathbf{x}_t$ | Unconditional score network $\mathbf{s}_{\boldsymbol{\theta}}(\mathbf{x}_t, \varnothing, t)$ |
| Classifier guidance [9] | Perturbed data $\mathbf{x}_t$ | Time-dependent classifier $h_t(\mathbf{x}_t, \mathbf{c})$ |
| Universal guidance [2] | Perturbed data $\mathbf{x}_t$ | Time-independent classifier $h(\bar{\mathbf{x}}_0(\mathbf{x}_t, \mathbf{c}))$ |
| DATE (ours) | Text embedding $\mathbf{c}_t$ | Time-independent classifier $h(\bar{\mathbf{x}}_0(\mathbf{x}_t, \mathbf{c}_t))$ |

## B.2 Guidance methods for diffusion models

Conditional diffusion models have been developed to generate samples that align with a given condition $y$. These models approximate the conditional score $\nabla_{\mathbf{x}_t} \log q_t(\mathbf{x}_t|y)$ by incorporating the condition as an additional input to the score network [9, 22]. In this paper, we represent this input as the condition embedding $\mathbf{c}$, which encodes the information of $y$.

To further improve conditional generation, diffusion models often incorporate a *guidance* term into the unconditional score function, as summarized in Table 6. Classifier guidance (CG) [9] introduces the gradient of a time-dependent classifier to encode conditional information.

$$\mathbf{s}_{\mathrm{CG}}(\mathbf{x}_t, \mathbf{c}, t) := \mathbf{s}_{\boldsymbol{\theta}}(\mathbf{x}_t, \varnothing, t) + w \nabla_{\mathbf{x}_t} \log h_t(\mathbf{x}_t, \mathbf{c}). \tag{40}$$

In contrast, classifier-free guidance (CFG) [19] eliminates the need for an explicit classifier by leveraging the difference between conditional and unconditional score estimates.

$$\mathbf{s}_{\mathrm{CFG}}(\mathbf{x}_t, \mathbf{c}, t) := \mathbf{s}_{\boldsymbol{\theta}}(\mathbf{x}_t, \varnothing, t) + w\big(\mathbf{s}_{\boldsymbol{\theta}}(\mathbf{x}_t, \mathbf{c}, t) - \mathbf{s}_{\boldsymbol{\theta}}(\mathbf{x}_t, \varnothing, t)\big). \tag{41}$$

Universal guidance (UG) [2] approximate the time-dependent guidance using a time-independent classifier, thereby avoiding time-dependent training.

$$\mathbf{s}_{\mathrm{UG}}(\mathbf{x}_t, \mathbf{c}, t) := \mathbf{s}_{\boldsymbol{\theta}}(\mathbf{x}_t, \varnothing, t) + w \nabla_{\mathbf{x}_t} \log h(\bar{\mathbf{x}}_0(\mathbf{x}_t, \mathbf{c})). \tag{42}$$

Similar to UG, our method also uses a time-independent classifier to derive guidance. However, instead of applying this guidance to the perturbed data, we use it to directly adjust the text embeddings, thus modifying the conditioning information as its source. The theoretical implications of this text embedding guidance are further analyzed in Theorem 2 from Section 3.3.

## B.3 Improving sampling process in fixed diffusion models

Recent studies have explored refining the sampling process in fixed diffusion models [23, 58, 38]. DG [23] adjusts the score network by incorporating an auxiliary term from a discriminator that differentiates real and generated samples, reducing network estimation errors during sampling process. Restart [58] alternates between reverse and forward steps at fixed time intervals. It first denoises samples with a deterministic sampler up to a predefined timestep, then injects noise to introduce stochasticity, and repeats this process to mitigate accumulated errors. DiffRS [38] evaluates sample quality at intermediate sampling steps using a discriminator, applies a rejection sampling scheme, and refines rejected samples by injecting instance-dependent stochastic noise.

Several studies have explored improving the sampling process in fixed diffusion models, specifically tailored for text-to-image diffusion models [4, 46, 35]. AnE [4] improves subject representation by updating the intermediate perturbed latent to maximize attention scores for subject tokens. SynGen [46] adjusts the intermediate perturbed latent to enforce linguistic binding between entities and their visual attributes by aligning the attention maps of paired tokens while differentiating the attention maps of unrelated word tokens. CONFORM [35] similarly updates the intermediate perturbed latent using contrastive loss on attention maps.

Unlike these methods, our approach does not require additional training of auxiliary components (e.g., a discriminator), updates the text embedding, and does not require additional information about the structure of the text prompt (e.g., binding token pairs). Moreover, our method can be integrated with existing approaches, as demonstrated in Section 4.3 with experiments using CONFORM.

# C  Additional experimental settings

## C.1  Experiments on COCO dataset

This subsection provides the experimental settings for Sections 4.1 and 4.2, where DATE is evaluated on the COCO validation set [30]. We use Stable Diffusion v1.5, pre-trained on LAION-5B [49], with a fixed CLIP ViT-L/14 text encoder [42] at a $512 \times 512$ resolution. We implement DATE on the Stable Diffusion pipeline with the Restart codebase,[3] built on Diffusers.[4] For EBCA [40], we use the official EBCA codebase,[5] which is also built on Diffusers, and its provided hyperparameters. We conducted most experiments on a single NVIDIA A100 GPU with CUDA 11.4, and some ablation studies were performed on a single Intel Gaudi v2 using SynapseAI 1.18.0. Our implementation is publicly available at https://github.com/aailab-kaist/DATE.

We use DDIM [51] with 50 sampling steps as the default sampler using classifier-free guidance [19], and experiments with the DDPM [18] sampler are provided in Appendix D.1. We set the guidance scale to 8 by default and provide results for various guidance scales in Appendix D.1. For DATE settings, unless otherwise specified, we set the text-conditioned evaluation function $h$ to CLIP score, computed using CLIP ViT-L/14 from the Hugging Face library. If we set $h$ to ImageReward, we compute ImageReward using the BLIP-based checkpoint from the official ImageReward codebase.[6] We set the scale hyperparameter $\rho$ to 0.5 and use the embedding from the previous update as the original text embedding $c_{\text{org}}$. For the ablation studies in Table 5, the text embedding is updated every 10% of the total sampling steps; and for the sensitivity analysis on $\rho$ in Figure 7, the text embedding is updated at every step.

We perform experiments across multiple backbones and samplers. For each backbone, we adopt the default sampler and configuration provided by the *diffusers* library. For PixArt-$\alpha$ [7] as the text encoder is used. Sampling follows the default setup, employing DPM-Solver [32] with 20 steps and a classifier-free guidance scale of 4.5. SD3 [11] incorporates CLIP-L/14, CLIP-bigG/14, and T5-XXL encoders, utilizing a flow-matching Euler sampler (28 steps) with a guidance scale of 7.0. FLUX [28] adopts a rectified flow transformer paired with CLIP-L/14 and T5-XXL text encoders, using a flow-matching Euler sampler (28 steps) and a guidance scale of 3.5. SDXL [41] relies on CLIP-L/14 and CLIP-bigG/14 as text encoders with a DDIM sampler, 50 steps, and a guidance scale of 5.0.

Implementations for PixArt-$\alpha$, SD3, FLUX, and SDXL are based respectively on the *PixArtAlphaPipeline*,[7] *StableDiffusion3Pipeline*,[8] *FluxPipeline*,[9] and *StableDiffusionXLPipeline*,[10] available in the Diffusers library.

Following the evaluation protocol of previous studies [39, 58, 38], we generate 5,000 images from randomly sampled captions in the COCO validation set. We fix the captions and random seed in all experiments. We evaluate text-to-image generation performance using zero-shot FID, CLIP score, and ImageReward. Zero-shot FID (Fréchet Inception Distance) [17, 45] measures the distributional similarity between real and generated images with the same text prompt in a feature space. Lower zero-shot FID values indicate that the generated images are more realistic and closer to the real image distribution. CLIP score [16] quantifies semantic alignment between a generated image and its text prompt by computing the cosine similarity between their embeddings in CLIP space [42]. A higher CLIP score suggests better text-image alignment. ImageReward [57] is a learned reward model trained on human preference data. Using a BLIP-based vision-language model [29], it evaluates text-image alignment and fidelity based on human judgment. A higher ImageReward score indicates that the generated image is more likely to be well aligned with human preferences.

---

[3] https://github.com/Newbeeer/diffusion_restart_sampling
[4] https://github.com/huggingface/diffusers
[5] https://github.com/EnergyAttention/Energy-Based-CrossAttention
[6] https://github.com/THUDM/ImageReward
[7] https://huggingface.co/docs/diffusers/main/en/api/pipelines/pixart
[8] https://huggingface.co/docs/diffusers/main/en/api/pipelines/stable_diffusion/stable_diffusion_3
[9] https://huggingface.co/docs/diffusers/main/en/api/pipelines/flux
[10] https://huggingface.co/docs/diffusers/main/en/api/pipelines/stable_diffusion/stable_diffusion_xl

Table 7: Prompt categories and examples on AnE dataset [4].

| Prompt Category | Template | Example | # of prompts |
|---|---|---|---|
| Animal-Animal | a [animalA] and a [animalB] | *a monkey and a frog* | 66 |
| Animal-Object | a [animal] and a [color][object] | *a monkey and a red car* | 144 |
| Object-Object | a [colorA][objectA] and a [colorB][objectB] | *a pink crown and a purple bow* | 66 |

## C.2 Multi-concept generation

We conduct our experiments on the Attend-and-Excite (AnE) dataset proposed by [4]. There are three types of prompts: 1) Animal-Animal: "a [animalA] and a [animalB]", 2) Animal–Object: "a [animal] and a [color][object]", and 3) Object–Object: "a [colorA][objectA] and a [colorB][objectB]". Detailed examples are provided in Table 7.

For baseline comparison, we evaluate our approach against base Stable Diffusion and CON-FORM [35]. We implement the CONFORM-based methods using its official codebase,[11] which is built on Diffusers. We use DDIM with 50 sampling steps using classifier-free guidance scale of 8. We set $h$ as CLIP score, the scale hyperparameter $\rho$ to 0.5, use the embedding from the previous update as the original text embedding $c_{\mathrm{org}}$, and text embedding is updated at every step.

We follow the quantitative evaluation protocol from previous studies [4, 35]. For each prompt, we generate images using 64 random seeds and evaluate performance based on text-image similarity and text-text similarity in CLIP space. *Full prompt similarity* measures the CLIP-based similarity between the entire prompt and the generated image. This metric measures the overall alignment, but it may not fully capture whether all concepts are represented. *Minimum object similarity* is computed by splitting the prompt into two sub-prompts and taking the lower CLIP similarity score between them, ensuring that even the least-represented concept is considered. For these similarities, we employ the CLIP ViT-B/16 model. The text embedding for each prompt is obtained by averaging the CLIP embeddings of 80 predefined prompt templates (e.g., "a good photo of a {*prompt*}.", "a photo of a clean {*prompt*}."). These similarities are then computed as the average similarity between this text embedding and the CLIP embeddings of the 64 generated images. For *text-caption similarity*, we generate captions for the 64 generated images using a pre-trained BLIP image-captioning model [29]. Then, we compute the CLIP similarity between the prompt's text embedding (obtained as described above) and the embeddings of the generated captions. The resulting similarity score is averaged across all generated images. To compute these metrics, we use the official implementation of AnE.[12]

## C.3 Text-guided image editing

We integrate DATE with DDPM inversion [21] on the ImageNet-R-TI2I dataset introduced in PnP [54]. DDPM inversion is a recent method that memorizes all latent vectors while tracing the inverse trajectory of a diffusion process. It generalizes DDIM inversion in the perspective of DDPM sampling framework. Our implementation is based on the official DDPM inversion codebase.[13]

For evaluation, we follow the parameter setting of DDPM inversion and vary the classifier-free guidance scale. Specifically, we use Stable Diffusion v1.4 with 100 sampling steps. DDPM inversion is evaluated with a guidance scale of $\{9, 12, 15, 18, 21\}$, while DATE is tested with a guidance scale of $\{6, 9, 12, 15\}$. We set $h$ as CLIP score, set $\rho$ to 0.5, initialize each step with the embedding from the previous one, and update the text embedding at every step. We report LPIPS [60] for perceptual similarity with the source image and CLIP score for alignment with the target prompt. LPIPS quantifies perceptual similarity using feature representations from a pre-trained VGG network [50], and CLIP score evaluates the cosine similarity between the target prompt embedding and the modified image embedding from a pre-trained CLIP model.

---

[11]https://github.com/gemlab-vt/CONFORM
[12]https://github.com/yuval-alaluf/Attend-and-Excite/tree/main/metrics
[13]https://github.com/inbarhub/DDPM_inversion

Table 8: Performance on the COCO validation set with Stable Diffusion v1.5 using the DDPM sampler with a classifier-free guidance scale of 8. Sampling steps are 50 unless otherwise specified. *Time* is the average sampling time (min.) for 64 samples, and *NFE* is the number of score network evaluations. **Bold** values indicate the best performance.

| Method | Time | NFE | Zero-shot FID↓ | CLIP score↑ | ImageReward↑ |
|---|---|---|---|---|---|
| Fixed text embedding | 5.76 | 100 | 21.94 | 0.3223 | 0.2567 |
| Fixed text embedding (steps=70) | 7.91 | 140 | 21.11 | 0.3212 | 0.2589 |
| EBCA [40] | 8.13 | 100 | 30.95 | 0.2851 | -0.2843 |
| **DATE (ours)** | | | | | |
|   10% update  with CLIP score | 7.90 | 105 | 20.78 | 0.3239 | 0.2630 |
|   all updates   with CLIP score | 24.21 | 150 | **20.68** | **0.3312** | 0.2712 |
|   10% update  with ImageReward | 7.90 | 105 | 21.14 | 0.3246 | 0.4913 |
|   all updates   with ImageReward | 24.21 | 150 | 21.33 | 0.3240 | **1.3262** |

Table 11: Computation time per sampling step for a batch size of 4.

| Operation | Time (sec.) |
|---|---|
| Uncond. and cond. score network evaluation (base sampling) | 0.33 |
| Text embedding update (including gradient computation) | 1.07 |
| Updated score network evaluation | 0.28 |

Table 12: GPU memory usage for a batch size of 4.

| Method | Memory (GB) |
|---|---|
| Fixed | 24.0 |
| DATE | 61.5 |

# D   Additional experimental results

## D.1   Additional experimental results on COCO dataset

**Other sampler and guidance scale**   Table 8 shows the experimental results of the baseline and DATE using the DDPM sampler, and Table 9 and Figure 15 show the results over different classifier-free guidance scales. Note that changing the classifier-free guidance scale does not affect the sampling time and NFE. We observe that DATE achieves performance improvements over the baseline in most metrics. These results demonstrate that our method consistently improves text-image alignment for generated images across various samplers and guidance scales.

**Additional experiment results on multi-objective optimization**   Table 10 reports results when using each metric, or their weighted combinations, is used as DATE's evaluation function. In most cases, DATE improves performance over the fixed embedding, regardless of whether a given metric is used as the evaluation function. One exception is when AS is included, where performance on other metrics often decreases, likely because AS, being independent of the text input, offers little synergy with semantic alignment metrics, as shown in Figure 4. Consistent with Figure 5b in the main text, the combined objective can yield higher values than using a single metric alone as indicated by the bold numbers in Table 10, demonstrating the synergistic potential of combining metrics during text embedding optimization.

**Computational costs**   Updating embeddings at each timestep increases the computational costs. As shown in the leftmost graph in Figure 9 of the main text, sampling time increases with update proportion. The overhead stems from extra score network evaluations for $x_0$, and gradient computations through $h$ and diffusion model, as mentioned in Section 3.4. A breakdown of the time required for each of these operations is provided in Table 11, and GPU memory usage is reported in Table 12.

Despite the added computational cost, Figure 14 shows that DATE consistently achieves better performance than fixed embeddings at comparable sampling times. We generate samples using text prompts of the COCO validation set with Stable Diffusion v1.5 using the DDIM sampler. Fixed embedding adjusts the number of sampling steps, and DATE adjusts both the number of sampling steps and embedding updates. DATE outperforms the fixed embedding on all evaluation metrics and sampling times. Notably, simply increasing the number of sampling steps in the fixed embedding setup yields only marginal improvements.

Table 9: Performance on the COCO validation set with Stable Diffusion v1.5, varying classifier-free guidance (CFG) scales. Sampling steps are 50 unless otherwise specified. For DATE, we apply a 10% update with CLIP score. **Bold** values indicate the best performance for each sampler and guidance scale.

| Sampler | CFG scale | Method | Zero-shot FID↓ | CLIP score↑ | ImageReward↑ |
|---|---|---|---|---|---|
| DDIM | 2 | Fixed text embedding | 15.90 | 0.2915 | -0.3616 |
| | | Fixed text embedding (steps=70) | 16.49 | 0.2905 | -0.3664 |
| | | EBCA [40] | 28.41 | 0.2492 | -0.9913 |
| | | **DATE (ours)** | **15.00** | **0.2959** | **-0.2838** |
| | 3 | Fixed text embedding | 14.04 | 0.3065 | -0.0947 |
| | | Fixed text embedding (steps=70) | 14.14 | 0.3056 | -0.0981 |
| | | EBCA [40] | 20.67 | 0.2710 | -0.6798 |
| | | **DATE (ours)** | **13.70** | **0.3089** | **-0.0451** |
| | 5 | Fixed text embedding | 16.14 | 0.3163 | 0.1165 |
| | | Fixed text embedding (steps=70) | 15.70 | 0.3155 | 0.1072 |
| | | EBCA [40] | 20.98 | 0.2842 | -0.4133 |
| | | **DATE (ours)** | **15.24** | **0.3182** | **0.1296** |
| | 8 | Fixed text embedding | 18.66 | 0.3204 | 0.2132 |
| | | Fixed text embedding (steps=70) | 18.27 | 0.3199 | 0.2137 |
| | | EBCA [40] | 25.85 | 0.2877 | -0.3128 |
| | | **DATE (ours)** | **17.90** | **0.3237** | **0.2364** |
| DDPM | 2 | Fixed text embedding | 14.07 | 0.3008 | -0.2173 |
| | | Fixed text embedding (steps=70) | **13.58** | 0.2999 | -0.2125 |
| | | EBCA [40] | 22.97 | 0.2629 | -0.8076 |
| | | **DATE (ours)** | 13.77 | **0.3033** | **-0.1864** |
| | 3 | Fixed text embedding | 15.17 | 0.3129 | 0.0315 |
| | | Fixed text embedding (steps=70) | **14.85** | 0.3120 | 0.0482 |
| | | EBCA [40] | 21.51 | 0.2779 | -0.5186 |
| | | **DATE (ours)** | 15.04 | **0.3158** | **0.0717** |
| | 5 | Fixed text embedding | 18.72 | 0.3199 | 0.1941 |
| | | Fixed text embedding (steps=70) | **18.27** | 0.3190 | 0.1974 |
| | | EBCA [40] | 25.07 | 0.2862 | -0.3178 |
| | | **DATE (ours)** | 18.32 | **0.3225** | **0.2051** |
| | 8 | Fixed text embedding | 21.94 | 0.3223 | 0.2567 |
| | | Fixed text embedding (steps=70) | 21.11 | 0.3212 | 0.2589 |
| | | EBCA [40] | 30.95 | 0.2851 | -0.2843 |
| | | **DATE (ours)** | **20.78** | **0.3239** | **0.2630** |

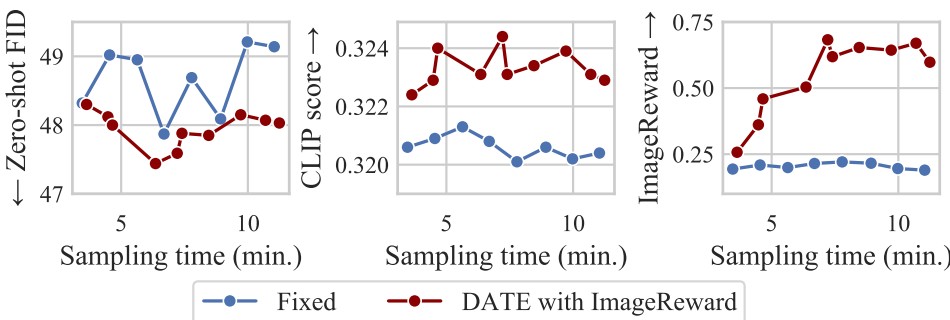

Figure 14: Performance comparison between fixed text embedding and DATE with ImageReward over different sampling times (minutes per 64 samples). FID values are computed using 1,000 samples, unlike the 5,000 samples used in the main text, which causes a scale discrepancy.

Table 10: Results on COCO with SD v1.5 (extension of Table 1 in the main text), including various combined evaluation functions. Columns 2–5 indicate which metrics are used in $h$. Higher and lower weights in columns 2—5 are denoted by ◉ and ○, respectively; a blank indicates the metric is not used. Bold numbers mark cases where combined metrics outperform the target single metric alone.

| Method | Evaluation function used in $h$ | | | | Metrics | | | | |
|---|---|---|---|---|---|---|---|---|---|
| | AS | CS | IR | PS | FID↓ | AS↑ | CS↑ | IR↑ | PS↑ |
| Fixed (50 steps) | | | | | 18.66 | 5.38 | 0.3204 | 0.2132 | 21.51 |
| Fixed (70 steps) | | | | | 18.27 | 5.37 | 0.3199 | 0.2137 | 21.50 |
| **DATE (50 steps, 10% update)** | | | | | | | | | |
| *with a single evaluation function* | | | | | | | | | |
| AS | ◉ | | | | 18.82 | 5.58 | 0.3169 | 0.1910 | 21.46 |
| CS | | ◉ | | | 17.90 | 5.35 | 0.3237 | 0.2364 | 21.53 |
| IR | | | ◉ | | 18.61 | 5.40 | 0.3224 | 0.4792 | 21.53 |
| PS | | | | ◉ | 18.49 | 5.42 | 0.3225 | 0.2745 | 21.93 |
| *with two combined evaluation functions* | | | | | | | | | |
| AS+CS | ◉ | ○ | | | 18.77 | 5.58 | 0.3171 | 0.1911 | 21.46 |
| | ○ | ◉ | | | 18.15 | 5.38 | 0.3219 | 0.2179 | 21.43 |
| AS+IR | ◉ | | ○ | | 18.90 | 5.57 | 0.3176 | 0.2428 | 21.48 |
| | ○ | | ◉ | | 18.15 | 5.43 | 0.3216 | 0.4575 | 21.53 |
| AS+PS | ◉ | | | ○ | 18.81 | 5.58 | 0.3175 | 0.2091 | 21.54 |
| | ○ | | | ◉ | 18.67 | 5.44 | 0.3219 | 0.2705 | 21.91 |
| CS+IR | | ◉ | ○ | | 17.94 | 5.39 | **0.3241** | 0.4430 | 21.52 |
| | | ○ | ◉ | | 18.04 | 5.40 | 0.3225 | 0.4756 | 21.53 |
| CS+PS | | ◉ | | ○ | 18.33 | 5.40 | **0.3241** | 0.2839 | 21.87 |
| | | ○ | | ◉ | 18.61 | 5.42 | 0.3224 | 0.2753 | 21.93 |
| IR+PS | | | ◉ | ○ | 18.15 | 5.41 | 0.3226 | 0.4774 | 21.63 |
| | | | ○ | ◉ | 18.54 | 5.42 | 0.3227 | 0.3126 | 21.93 |
| *with three combined evaluation functions* | | | | | | | | | |
| AS+CS+IR | ◉ | ○ | ○ | | 18.87 | 5.57 | 0.3179 | 0.2487 | 21.48 |
| | ○ | ◉ | ○ | | 18.50 | 5.46 | 0.3208 | 0.3332 | 21.50 |
| | ○ | ○ | ◉ | | 18.07 | 5.43 | 0.3219 | 0.4557 | 21.53 |
| AS+CS+PS | ◉ | ○ | | ○ | 18.96 | 5.57 | 0.3179 | 0.2092 | 21.55 |
| | ○ | ◉ | | ○ | 18.62 | 5.47 | 0.3211 | 0.2450 | 21.69 |
| | ○ | ○ | | ◉ | 18.69 | 5.44 | 0.3221 | 0.2705 | 21.91 |
| AS+IR+PS | ◉ | | ○ | ○ | 18.93 | 5.57 | 0.3186 | 0.2602 | 21.56 |
| | ○ | | ◉ | ○ | 18.33 | 5.44 | 0.3222 | 0.4599 | 21.61 |
| | ○ | | ○ | ◉ | 18.72 | 5.44 | 0.3224 | 0.3071 | 21.91 |
| CS+IR+PS | | ◉ | ○ | ○ | 18.32 | 5.41 | **0.3244** | 0.4192 | 21.81 |
| | | ○ | ◉ | ○ | 18.16 | 5.41 | 0.3229 | **0.4812** | 21.63 |
| | | ○ | ○ | ◉ | 18.68 | 5.42 | 0.3228 | 0.3138 | 21.92 |

**Statistical significance**   To assess the statistical significance of the CLIP score improvements introduced by using ImageReward as $h$, we conduct a paired t-test. Comparing samples generated with fixed embeddings (sampling time = 6.34 minutes) and DATE (sampling time = 6.03 minutes), we obtain a p-value of 0.00056, indicating a statistically significant improvement at comparable sampling costs.

## D.2  Additional analysis of DATE

**Multiple text embedding updates**   We hypothesize that performing multiple text embedding updates per sampling step can expand the search space beyond the initial $\rho$-ball, potentially leading to improved performance. Figure 16 indicates that the CLIP score generally increases with more update iterations. However, each additional iteration incurs extra forward and backward passes, resulting in a linear increase in sampling time.

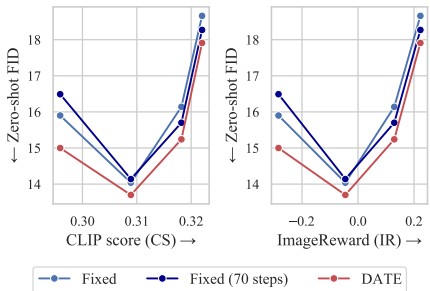

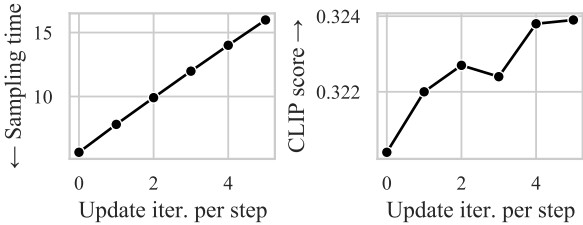

Figure 15: Performance between FID and text-image alignment metrics varying the classifier-free guidance scale with the DDIM sampler.

Figure 16: Sensitivity analysis on the number of update iterations per sampling step with a 50-step DDIM sampler and a classifier-free guidance scale of 8. For DATE, we apply a 10% update with CLIP score. When the number of iterations is 0, it is identical to fixed embedding.

Text prompt: *A bearded man in a wetsuit holding a surfboard.*

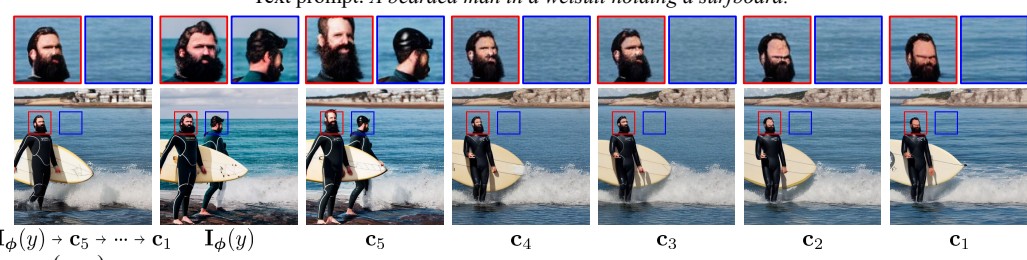

Figure 17: Generated images of DATE and various fixed text embeddings. The bottom image in each column is the generated image, while the two images above it are zoomed-in views of the boxed regions in the generated image. $\mathbf{I}_\phi(y)$ represents the text embedding from the original text encoder, while $\mathbf{c}_5, \mathbf{c}_4, \ldots, \mathbf{c}_1$ represent the text embeddings updated during the intermediate sampling steps of DATE, with larger indices indicating earlier stages of sampling. The leftmost image is generated using DATE with dynamic updates, while the remaining images are generated with fixed text embeddings.

**Time-adaptive text embedding**   To analyze the updated text embeddings of DATE, we inject the each updated embedding into the entire sampling process. Figure 17 shows the generated images from DATE and several fixed text embeddings. With the embedding obtained after the middle sampling step ($\mathbf{c}_4$), we observe that the information '*two men*' is transformed into '*a man*' in the text embedding. Furthermore, when using the updated text embeddings at later sampling steps ($\mathbf{c}_2, \mathbf{c}_1$), we observe that the face region of the generated image appears distorted. This suggests that the final updated text embedding is not necessarily globally optimal, and that an appropriate text embedding may exist at each diffusion timestep.

### D.3   Additional results for applications

**Multi-concept generation**   In addition to the main evaluation metrics, we also assess model performance using the TIFA score [20], a recently proposed metric designed to measure the faithfulness of generated images to their textual prompts. TIFA leverages a Visual Question Answering (VQA) model to quantify alignment between image content and prompt, providing an evaluation that is independent of CLIP.

Table 13 presents the full evaluation results across all prompt types in the multi-concept generation setting. We also include results for DATE with ImageReward as $h$. DATE consistently improves the TIFA score across all tested cases. Furthermore, when using ImageReward as $h$, DATE continues to outperform the baseline across all evaluation metrics, highlighting its robustness and effectiveness regardless of the chosen evaluation function. In addition, Figure 18 shows more generated images for various prompts from the AnE dataset. These results demonstrate that DATE effectively applies to multi-concept generation methods, enabling the generation of images that accurately reflect the given concepts.

Table 13: Performance comparison for multi-concept generation on the AnE dataset, compared across Stable Diffusion, DATE with ImageReward, and DATE with CLIP score.

| Prompt type | Method | Full prompt | Min. object | Text-caption | TIFA score |
|---|---|---|---|---|---|
| Animal-Animal | Stable Diffusion | 0.3123 | 0.2174 | 0.7677 | 0.6847 |
| | + DATE (ImageReward) | 0.3219 | 0.2371 | 0.7858 | 0.7948 |
| | + DATE (CLIP score) | **0.3282** | **0.2398** | **0.7888** | **0.8159** |
| Animal-Object | Stable Diffusion | 0.3443 | 0.2480 | 0.7925 | 0.8223 |
| | + DATE (ImageReward) | 0.3454 | 0.2512 | **0.8009** | **0.8486** |
| | + DATE (CLIP score) | **0.3530** | **0.2568** | **0.8009** | 0.8420 |
| Object-Object | Stable Diffusion | 0.3377 | 0.2404 | 0.7684 | 0.6402 |
| | + DATE (ImageReward) | 0.3391 | 0.2454 | 0.7706 | **0.6910** |
| | + DATE (CLIP score) | **0.3503** | **0.2544** | **0.7728** | 0.6643 |

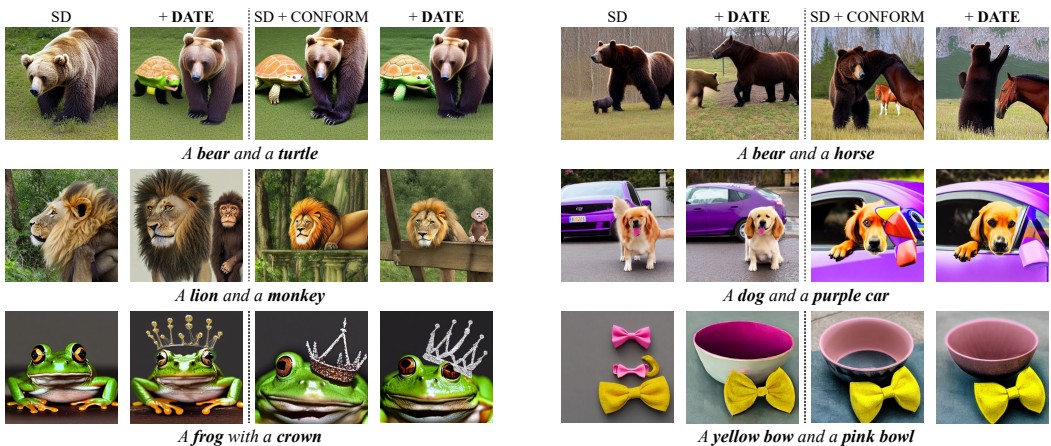

Figure 18: Additional generated images on the AnE dataset for multi-concept generation.

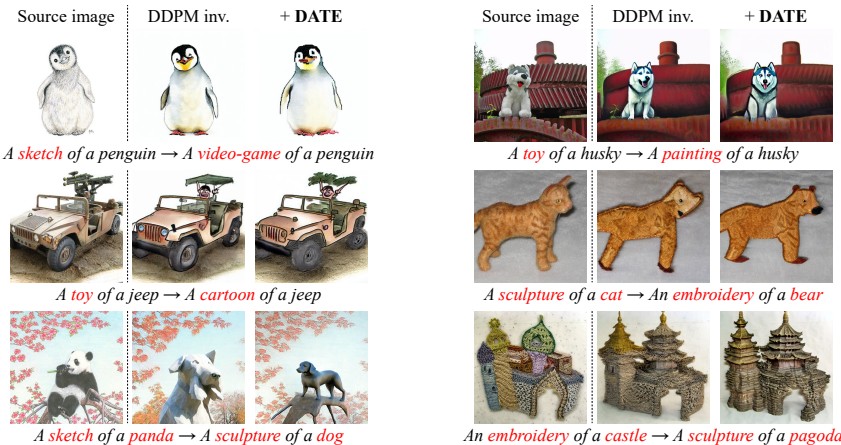

Figure 19: Additional examples of edited images on the ImageNet-R-TI2I dataset for text-guided image editing.

**Text-guided image editing** Figure 19 shows several examples of text-guided image editing using DDPM inversion with and without DATE applied to the source images. For each method, we present the processed image obtained with the hyperparameter that makes LPIPS less than 0.25. These examples demonstrate that applying DATE to DDPM inversion improves text-guided image editing by better preserving the structure of the source image while improving alignment with the target prompt.

## E   Limitation and broader impact

**Limitation**   One primary limitation of our approach is the additional computational overhead introduced by updating text embeddings during the sampling process. These updates require gradient computations, resulting in increased sampling time and memory usage. However, our experiments demonstrate that DATE outperforms fixed embeddings under equivalent sampling-time budgets, suggesting a favorable trade-off between efficiency and effectiveness. Nevertheless, repeated gradient computations present challenges in terms of memory and computing efficiency, especially in resource-constrained settings. Developing memory-efficient techniques for gradient updates is an important direction for future work.

Another potential limitation is the dependence on the evaluation function. While we explore the use of multi-objective evaluation functions and provide empirical evidence that DATE does not overfit the evaluation function itself, the overall generation quality can still be influenced by the design and reliability of the evaluation function. This highlights a broader challenge in text-to-image generation: the field continues to lack fully reliable, general-purpose evaluation metrics. Continued research on evaluation protocols and their integration with guidance mechanisms is crucial for advancing robust and generalizable generation frameworks.

**Broader impact**   Improving conditional embeddings in diffusion-based generative models remains an underexplored area, despite being a key component of conditional generation. Our work addresses this gap by proposing a general and effective method to refine text embeddings during sampling, thereby enhancing alignment between the prompt and the generated image. A significant advantage of our approach is that it operates without requiring any additional model training and is agnostic to the backbone model and sampler. This makes it readily applicable to a wide range of text-to-image generation systems.

However, using external modules introduces potential vectors for misuse. For example, adversarial manipulation of these components could compromise model safety and lead to unintended or harmful outputs. To mitigate these risks, appropriate safeguards could be incorporated into the evaluation functions and sampling process. Responsible deployment of such systems should account for these concerns.

## F   License information

Our implementation will be publicly released under standard community licenses. In addition, we provide the license information for the datasets and models used in this paper:

**SD v1.5:**  https://huggingface.co/spaces/CompVis/stable-diffusion-license

**PixArt-$\alpha$:**  https://github.com/PixArt-alpha/PixArt-alpha/blob/master/LICENSE

**CLIP:**  https://github.com/openai/CLIP/blob/main/LICENSE

**ImageReward:**
      https://github.com/THUDM/ImageReward/blob/main/LICENSE

**TIFA:**  https://github.com/Yushi-Hu/tifa/blob/main/LICENSE

**COCO:**  https://cocodataset.org/#termsofuse

**AnE:**  https://github.com/yuval-alaluf/Attend-and-Excite/blob/main/
      LICENSE

**ImageNet-R-TI2I:**
      https://github.com/MichalGeyer/plug-and-play

**Restart:**  https://github.com/Newbeeer/diffusion_restart_sampling

**EBCA:** https://github.com/EnergyAttention/Energy-Based-CrossAttention

**CONFORM:**
https://github.com/gemlab-vt/CONFORM/blob/main/LICENSE

**DDPM Inversion:**
https://github.com/inbarhub/DDPM_inversion/blob/main/LICENSE

