# OpenReview forum: "Diffusion Adaptive Text Embedding for Text-to-Image Diffusion Models"
_NeurIPS.cc/2025/Conference — NeurIPS 2025 poster_

### Official Review · Reviewer_TJGJ · 2025-06-24

**Clarity:** 3
**Significance:** 2
**Originality:** 3
**Rating:** 4
**Confidence:** 3

**Summary:**

The paper introduces Diffusion Adaptive Text Embedding (DATE), a method that improves text-to-image diffusion models by dynamically updating text embeddings at each diffusion timestep. Unlike traditional approaches with fixed embeddings, DATE refines embeddings based on intermediate image data during sampling, enhancing text-image alignment without additional model training. Theoretical and empirical results show that DATE maintains generative quality while improving performance in tasks like multi-concept generation and text-guided editing.

**Questions:**

Can the method be transferred to other stronger base models like SD-XL, FLUX or other DiT-based image generation models? If so, I would like to see how actually it improves.

**Ethical Concerns:**

["NO or VERY MINOR ethics concerns only"]

**Final Justification:**

My concerns are addressed, hence I maintain my rating.

**Limitations:**

Please see the weaknesses.

**Paper Formatting Concerns:**

No.

**Quality:**

3

**Strengths And Weaknesses:**

Pros:
1. The paper presents an interesting formulation of the problem and offers a unique perspective.
2. The overall writing is clear and easy to follow.
3. The results seem promising in some cases.

Cons:
1. The results presented in this paper are limited to some easy cases. As the authors mainly focus on optimizing the text embedding, they should put more attention on complex, long texts to show the improvement they claim to achieve.
2. Additionally, there are actually many powerful image generation models like GPT-4o, Gemini, FLUX that can achieve good image generation quality while accurately following the text prompts. The authors should compare with them (although some are not open-sourced, they can be inferenced for comparison).
3. They authors use CLIP Score or ImageReward as the optimization function but also use them as the evalution may lead the optimization into sub-optimal and overfitiing. The authors should explain its validity.

---

> ### Author Rebuttal · Authors · 2025-07-31
>
> We are grateful for the reviewers' thoughtful evaluations and detailed comments. Our responses to the raised points are presented below.
> ***
> > ### **W1: [Long and complex prompts]**
>
> Text encoders in diffusion models are often limited by a fixed token length, which can pose challenges for handling long and complex prompts. For example, many diffusion models using CLIP encoders are restricted to 77 tokens, and the tokens beyond this limit are typically truncated and ignored. Since DATE provides guidance through an external function and does not rely solely on the internal text encoder. This allows DATE to influence the generation process beyond the model’s fixed token-length limitation, making it a promising solution for handling long and complex prompts.
>
> To investigate this, we perform experiments on the LongAlign (Liu et al., 2025), which contains prompts of substantial length and complexity. We sample 300 prompts and use ImageReward (20% updates) as the evaluation function. For ImageReward, each long prompt is split into individual sentences, and at each update step, one sentence is randomly selected to compute the score and guide the embedding update. We use Denscore (Liu et al., 2025) for the metric.
>
> **[Table: Results for long prompts on LongAlign dataset (SD v1.5)]**
> |Method|Denscore↑|
> |---|---|
> |Fixed|0.2034|
> |**DATE**|**0.2042**|
>
> **[Table: Average Denscore by sentence position]**
> ||S1|S2|S3|S4|S5|S6+|
> |---|---|---|---|---|---|---|
> |Fixed|**0.2597**|**0.2478**|**0.2234**|0.1961|0.1678|0.1764|
> |**DATE**|0.2561|0.2465|0.2230|**0.1991**|**0.1729**|**0.1799**|
>
> These results show that DATE improves Denscore compared to the baseline. Notably, the improvement is more significant for sentences appearing later in the prompt, suggesting that DATE can inject additional semantic guidance in regions where the base model typically underperforms due to token truncation. These results suggest that DATE can serve as an effective mechanism for injecting auxiliary information into diffusion models, especially in scenarios where prompt length exceeds the native capacity of the text encoder.
>
> (Liu et al., 2025) Improving Long-Text Alignment for Text-to-Image Diffusion Models, ICLR 2025.
> ***
> > ### **W2 & Q: [More powerful generation models]**
>
> Thank you for the suggestion. First, we would like to emphasize that DATE is an inference-time method designed to improve existing image generation models by guiding the sampling process. It can be applied on top of powerful base models to further improve their performance without requiring retraining or access to large-scale paired data. That said, we do not claim that our approach outperforms methods that rely on extensive training with massive datasets and computing resources. Rather, our goal is to provide an adaptive improvement that can complement such powerful image generation models, even when full retraining is impractical or infeasible.
>
> While DATE does not modify model weights, it does require access to model weights due to computation of gradients, which makes it challenging to apply to closed-source models such as GPT-4o or Gemini.
> On the other hand, for open-source models, DATE is readily applicable. As shown in Table 1 of the main text, DATE consistently improves performance across all metrics when applied to PixArt-alpha, which is a DiT-based image generation model.
> To further evaluate DATE’s applicability to a broader range of strong base models, we additionally apply DATE to FLUX (guided-distilled version), SD-XL and SD3.  For each model, we follow the default configurations provided in the HuggingFace diffusers library and use ImageReward as the evaluation function.
>
> **[Table: Results on COCO with recent stronger base models]**
> |Backbone|Method| FID↓|CLIP score↑|ImageReward↑|
> |---|---|---|---|---|
> |SD3| Fixed|26.00|0.3337|1.0018|
> ||**DATE**|26.00|**0.3340**|**1.0457**|
> |FLUX|Fixed|29.59|0.3257|0.9634|
> ||**DATE**|**29.41**|**0.3283**|**0.9768**|
> |SDXL|Fixed|18.27|0.3368| 0.7284|
> ||**DATE**|**18.03**|**0.3382**|**0.9096**|
>
> The results above follow the evaluation setup in Table 1 of the main text. Across all modern architectures, DATE consistently improves performance on all metrics compared to the fixed text embedding baseline. These results suggest that DATE can be transferred to a range of stronger base models and consistently enhance their generation quality.
>
> ***
> > ### **W3: [Reliance on evaluation function]**
>
> We acknowledge that DATE depends on the reward function, as discussed in Appendix E. Here, we offer theoretical, empirical, and practical perspectives.
>
> **(Theoretical perspective)**
>
> Theorem 2 shows DATE aligns text embeddings with the reward function while also regularizing with the model's text-conditional likelihood. This means that DATE does not blindly overfit to the reward function but instead applies regularized guidance informed by the underlying diffusion model. This effect can reduce the risk of overfitting to the proxy metrics.
>
> **(Empirical perspective)**
>
> Empirically, as shown in Table 1 of the main text, DATE improves performance on both the reward function used for optimization and other metrics not used during optimization (e.g., FID). To further validate this generalization, we conduct additional experiments using PickScore (Kirstain et al., 2023) as both an optimization objective and an external evaluation metric. As shown in the table, DATE consistently improves performance across all metrics. This suggests that DATE’s improvements reflect broader enhancements in generation quality. Ongoing research continues to explore more faithful evaluation metrics for text-to-image generation, and DATE can be readily combined with such advances.
>
> **[Table: Results on COCO with SD v1.5 of Table 1 in the main text, including PickScore.]**
> ||Time|FID↓|CLIP score↑|ImageReward↑|PickScore↑|
> |---|---|---|---|---|---|
> |Fixed (50 steps)|5.64|18.66|0.3204|0.2132|16.8590|
> |Fixed (70 steps)|7.87|18.27|0.3199|0.2137|16.8646|
> |**DATE** (50 steps, 10% update)|
> |&nbsp;with CLIP score|7.82|**17.90**|**0.3237**|0.2364|16.8728|
> |&nbsp;with ImageReward|7.82|18.61|0.3224| **0.4792**|16.8750|
> |&nbsp;with PickScore|8.23|18.49|0.3225|0.2745|**16.8761**|
>
> (Kirstain et al., 2023) Pick-a-Pic: An Open Dataset of User Preferences for Text-to-Image Generation, NeurIPS 2023.
>
> **(Practical perspective)**
>
> Our method also supports multi-objective optimization, which can further mitigate overfitting to a single metric. In Figure 4 of the main text, we show results using a combined objective of CLIP score and ImageReward. Interestingly, this combination outperforms the use of each metric individually. This result highlights the potential of our method to leverage synergy between multiple objectives for more robust guidance.
>
> In summary, while reward function dependency is an important consideration, DATE incorporates both theoretical regularization and practical flexibility to address it effectively.

---

> > ### Comment · Reviewer_TJGJ · 2025-08-04
> > **Official Reply by Reviewer TJGJ**
> >
> > Thanks for the authors' response, which has addressed most of my concerns. Hence, I tend to accept this paper.

---

> > > ### Author Response · Authors · 2025-08-04
> > >
> > > Thank you for the reviewer’s response and thoughtful consideration. We’re encouraged to hear that our clarifications helped address the reviewer's concerns. We appreciate the time and effort dedicated to reviewing our work.

---

### Official Review · Reviewer_dcFc · 2025-06-28

**Clarity:** 3
**Significance:** 3
**Originality:** 3
**Rating:** 4
**Confidence:** 4

**Summary:**

This paper proposes Diffusion Adaptive Text Embedding (DATE), a novel method for improving text-to-image diffusion models by dynamically updating text embeddings during the sampling process.  Experimental results across several tasks, including t2i generation, multi-concept generation, and text-guided image editing.

**Questions:**

See the Weakness part.

**Ethical Concerns:**

["NO or VERY MINOR ethics concerns only"]

**Final Justification:**

The author addressed my concerns on reliance on evaluation function, multiple text embeddings, and \etc. They also provided a detailed analysis of the reward model, which is very important in fine-tuning Text-to-Image Diffusion Models. Therefore, I increase my rating from borderline reject to borderline accept.

**Limitations:**

yes.

**Paper Formatting Concerns:**

NA.

**Quality:**

3

**Strengths And Weaknesses:**

Thanks to the authors for their efforts in this work.

**Strengths**
This manuscript is well-presented, with comprehensive experimental results and solid theoretical analysis. The idea that updates the textual embedding with an optimization strategy is novel. The experiments are conducted across different network architectures (e.g., Stable Diffusion v1.5 and PixArt-α), and include detailed ablation studies. In addition, the authors provide a clear theoretical formulation of their method, including optimization objectives and update strategies.

**Weaknesses**
1. Despite the effective optimization formulation, the proposed framework relies on the evaluation function $h$, such as CLIP score,  which is known to exhibit biases [1]. Such reliance may limit the robustness or generalizability of the method under varying evaluation metrics. while it also has a trade-off of the Clip score and ImageReward.
2. Some arguments are overclaims; e.g., the related work section (lines 70–72) lacks discussion of relevant approaches such as SD3 [2], which utilizes multiple text embeddings.
3. There is also some related work on text embedding tuning [3], text-to-image generation based on the evaluation function [4], which needs to be discussed.
4. A key concern is that, although the experimental results demonstrate that the proposed embedding tuning improves performance somehow, it alters the embedding space of the original input text. In other words, it may deviate from the user's original intent in generation.

$\small{\text{ [1] Ruiz, Nataniel, et al. "Dreambooth: Fine tuning text-to-image diffusion models for subject-driven generation." Proceedings of the IEEE/CVF conference on computer vision and pattern recognition. 2023.}}$
$\small{\text{ [2] Esser, Patrick, et al. "Scaling rectified flow transformers for high-resolution image synthesis." Forty-first international conference on machine learning. 2024.}}$
$\small{\text{ [3] Gal, Rinon, et al. "An image is worth one word: Personalizing text-to-image generation using textual inversion." arXiv preprint arXiv:2208.01618 (2022).}}$
$\small{\text{ [4] Fan, Ying, et al. "Dpok: Reinforcement learning for fine-tuning text-to-image diffusion models." Advances in Neural Information Processing Systems 36 (2023): 79858-79885.}}$

---

> ### Author Rebuttal · Authors · 2025-07-31
>
> We thank the reviewer for thoughtful comments. We address the questions in the following response.
> ***
> > ### **W1: [Reliance on evaluation function]**
>
> We acknowledge that DATE depends on the reward function, as discussed in Appendix E. Here, we offer theoretical, empirical, and practical perspectives.
>
> **(Theoretical perspective)**
>
> Theorem 2 shows DATE aligns text embeddings with the reward function while also regularizing with the model's text-conditional likelihood. This means that DATE does not blindly overfit to the reward function but instead applies regularized guidance informed by the underlying diffusion model. This effect can reduce the risk of overfitting to the proxy metrics.
>
> **(Empirical perspective)**
>
> Empirically, as shown in Table 1 of the main text, DATE improves performance on both the reward function used for optimization and other metrics not used during optimization (e.g., FID). To further validate this generalization, we conduct additional experiments using PickScore (Kirstain et al., 2023) as both an optimization objective and an external evaluation metric. As shown in the table, DATE consistently improves performance across all metrics. This suggests that DATE’s improvements reflect broader enhancements in generation quality. Ongoing research continues to explore more faithful evaluation metrics for text-to-image generation, and DATE can be readily combined with such advances.
>
> **[Table: Results on COCO with SD v1.5 of Table 1 in the main text, including PickScore.]**
> ||Time|FID↓|CLIP score↑|ImageReward↑|PickScore↑|
> |---|---|---|---|---|---|
> |Fixed (50 steps)|5.64|18.66|0.3204|0.2132|16.8590|
> |Fixed (70 steps)|7.87|18.27|0.3199|0.2137|16.8646|
> |**DATE** (50 steps, 10% update)|
> |&nbsp;with CLIP score|7.82|**17.90**|**0.3237**|0.2364|16.8728|
> |&nbsp;with ImageReward|7.82|18.61|0.3224| **0.4792**|16.8750|
> |&nbsp;with PickScore|8.23|18.49|0.3225|0.2745|**16.8761**|
>
> (Kirstain et al., 2023) Pick-a-Pic: An Open Dataset of User Preferences for Text-to-Image Generation, NeurIPS 2023.
>
> **(Practical perspective)**
>
> Our method also supports multi-objective optimization, which can further mitigate overfitting to a single metric. In Figure 4 of the main text, we show results using a combined objective of CLIP score and ImageReward. Interestingly, this combination outperforms the use of each metric individually. This result highlights the potential of our method to leverage synergy between multiple objectives for more robust guidance.
>
> In summary, while reward function dependency is an important consideration, DATE incorporates both theoretical regularization and practical flexibility to address it effectively.
> ***
> > ### **W2: [Multiple text embeddings]**
>
> Thank you for the thoughtful feedback. As we present dynamically adapted text embedding to the diffusion process, our method (DATE) is distinct from a static set of multiple text embeddings, such as SD3.
>
> Regarding the related work section, we agree that methods like SD3 are relevant to the goal of improving text embeddings, and we will include a discussion of such methods in the revised version to provide a more comprehensive context.
> That said, our intention in lines 70-72 was to highlight a specific distinction: while methods like SD3 improve text embeddings (e.g., by using multiple text encoders), they generally do not adapt these embeddings dynamically across diffusion timesteps. In contrast, DATE is motivated by this limitation and proposes an inference-time mechanism for updating the text embedding dynamically over timesteps.
> Importantly, this difference makes DATE complementary to approaches like SD3. We demonstrate that DATE can be applied on top of SD3, SD-XL, and FLUX, all of which use multiple text embedding. For each model, we follow the default configurations provided in the HuggingFace diffusers library and use ImageReward as the evaluation function.
>
> **[Table: Results on COCO with recent stronger base models]**
> |Backbone|Method| FID↓|CLIP score↑|ImageReward↑|
> |---|---|---|---|---|
> |SD3| Fixed|26.00|0.3337|1.0018|
> ||**DATE**|26.00|**0.3340**|**1.0457**|
> |FLUX|Fixed|29.59|0.3257|0.9634|
> ||**DATE**|**29.41**|**0.3283**|**0.9768**|
> |SDXL|Fixed|18.27|0.3368| 0.7284|
> ||**DATE**|**18.03**|**0.3382**|**0.9096**|
>
> The results above follow the evaluation setup in Table 1 of the main text. Across all modern architectures, DATE consistently improves performance on all metrics compared to the fixed text embedding baseline. These results demonstrates that DATE can be effectively integrated with models that utilize multiple text embeddings, providing a scalable and complementary improvement.
> ***
> > ### **W3: [Related works]**
>
> We agree that a more comprehensive comparison including methods for prompt optimization, text embedding tuning, and reward-based fine-tuning would strengthen the paper. In the revised version, we include a detailed discussion and additional experiments covering the following areas, as below.
>
> As discussed in Section 2.2 of the main text, methods for refining text-to-image generation can generally target one of the following: model parameters, perturbed data, or text embeddings. The work in [4], as well as (Black et al., 2024) cited in our paper, falls into the first category. These approaches fine-tune the model parameters using a reward function. However, they require significant training time and computational resources.
>
> Prompt-level optimization targets the input text. Several learning-based approaches (Hao et al., 2023, Mo et al., 2024) have explored prompt optimization using reward models and reinforcement learning. These methods train a language model to generate prompts optimized for a diffusion model based on reward signals. However, such methods typically require significant training time and computational resources. Additionally, if the diffusion backbone or reward function changes, the model could be retrained. In contrast, DATE is training-free and operates at inference time. Moreover, these learning-based previous works could be used in conjunction with DATE during test-time for complementary benefits.
>
> There is also related work focusing on tuning text embeddings directly. For example, [3] learns a special token embedding using a small number of personalized images, which is then appended to prompts for personalized image generation. Similarly, (Um et al., 2025) introduce a method for generating minority instances by learning a special token embedding, further extending it to be time-dependent during sampling. While [3] and (Um et al., 2025) focus on specific tasks, DATE proposes a general-purpose framework for improving text-to-image alignment. Additionally, a key distinction lies in the granularity of updates: [3] and (Um et al., 2025) optimize a special token embedding, whereas DATE updates the entire text embedding vector.
>
> We conduct additional experiments by combining DATE with Promptist (Hao et al., 2023), a representative prompt tuning method. We use ImageReward as the evaluation function for DATE.
>
> **[Table: Results with the prompt tuning, Promptist (Hao et al., 2023). FID values may differ in scale due to the smaller sample size (1k).]**
> |Method| FID-1K↓ |CLIP score↑ | ImageReward↑ | Aesthetic score↑ |
> |---|---|---|---|---|
> |Fixed|48.81|0.3206|0.1801|5.40|
> |**DATE**|48.09|0.3233|0.4680|5.43|
> |Promptist|57.75|0.2785|-0.0380|6.23|
> |Promptist+**DATE**|56.23|0.2827|0.2471|6.20|
>
> Promptist focuses on improving aesthetic quality (Aesthetic score), often at the cost of other metrics. In contrast, DATE improves all metrics, including Aesthetic score, which is not used in its objective, indicating good generalization. When combined with Promptist, DATE helps recover image-text alignment and overall quality while maintaining high aesthetic quality. This suggests DATE can complement prompt tuning methods.
>
> (Black et al., 2024) Training Diffusion Models with Reinforcement Learning, ICLR 2024.
> (Hao et al., 2023) Optimizing Prompts for Text-to-Image Generation, NeurIPS 2023.
> (Mo et al., 2024) Dynamic Prompt Optimizing for Text-to-Image Generation, CVPR 2024.
> (Um et al., 2025) Minority-Focused Text-to-Image Generation via Prompt Optimization, CVPR 2025.
> ***
> > ### **W4: [Deviation from the original intent]**
>
> We acknowledge the potential risk that adapting the text embedding might deviate from the user’s original intent. However, we believe DATE can actually help better reflect that intent through a feedback-driven refinement process.
> Our method updates the text embedding based on **the expected image output $\bar{x}_0$** generated from the current embedding, guided by an external evaluation function. This allows the embedding to shift in a direction that better aligns with the user’s intent as captured by the evaluation signal. In contrast to fixed text embeddings, which rely solely on the static encoding of the input text, DATE introduces a feedback loop between the intermediate generated image and the text embedding, making it more adaptive to the desired semantics.
>
> Moreover, Theorem 2 provides a theoretical justification showing that DATE updates the text embedding by aligning with the evaluation function while also accounting for the original diffusion model’s text-conditional likelihood. This regularization effect helps prevent overfitting to the evaluation function or losing alignment with the original prompt semantics.
>
> That said, we agree that since the base diffusion model was not trained with dynamically updated text embeddings, there may be a training-inference gap. To address this, we explicitly introduce a constraint on the embedding update: $||c-c_{org}||_2 \leq \rho$ in Eq. (7). This constraint allows us to balance preserving the original text semantics while adapting to the evaluation feedback. As shown in the sensitivity analysis in Figure 6 of the main text, performance improves when $\rho$ is set appropriately, supporting the effectiveness of this trade-off.

---

> > ### Comment · Reviewer_dcFc · 2025-08-05
> >
> > Thank you for your efforts in the rebuttal. Most of my concerns have been addressed quite well, and I am willing to raise my rating. Additionally, I would appreciate it if you could provide more analysis regarding the different reward models, *e.g.*, as shown in [Table: Results on COCO with SD v1.5 of Table 1 in the main text, including PickScore.], how different reward models (ImageReward *vs* PickScore) may have varying strengths, such as emphasizing semantics versus image fidelity.

---

> > > ### Comment · Reviewer_dcFc · 2025-08-08
> > >
> > > Hi, as the author–reviewer discussion period is coming to an end, do you have any updated discussions?

---

> > > > ### Author Response · Authors · 2025-08-08
> > > >
> > > > Sorry for the delayed response. We needed additional time to run the extra experiments for the reward model analysis. We are pleased that most of the concerns have been addressed through the rebuttal, and as suggested, we additionally conducted a detailed analysis of different reward models.
> > > >
> > > > Following the approach in Figure 4 of the main paper, we (1) compare pairwise correlations among reward models (Table A), and (2) evaluate DATE using single or combined models as the evaluation function (Table B). The reward models considered are **Aesthetic Score (AS)** for image fidelity and aesthetic quality, **CLIP Score (CS)** for semantic alignment, and two human-preference-based metrics, **ImageReward (IR)** and **PickScore (PS)**. Each produces a score for each generated image, enabling both correlation analysis and direct use as DATE’s evaluation function. Note that FID is excluded since it is a distribution-level metric without per-instance scores.
> > > >
> > > > **[Correlation analysis]**
> > > >
> > > > Table A shows the Pearson correlation coefficients computed over 1,000 Stable Diffusion samples. AS and CS have a low correlation (0.085), indicating that aesthetic quality and text-image alignment are largely independent. IR and PS correlate moderately with CS (IR: 0.502, PS: 0.465) and weakly with AS (IR: 0.180, PS: 0.252), suggesting they capture a blend of fidelity and semantic factors. IR and PS correlate with each other (0.437) but not strongly, likely due to differences in training data and model architecture. PS is more sensitive to aesthetics, whereas IR is more sensitive with semantics.
> > > >
> > > > **[Table A: Correlation matrix among metrics.]**
> > > > |||AS|CS|IR|PS|
> > > > |---|---|---|---|---|---|
> > > > |Fidelity|**AS**|||||
> > > > |Semantic alignment|**CS**|0.085||||
> > > > |Human preference|**IR**|0.180|0.502|||
> > > > |Human preference|**PS**|0.252|0.465|0.437||
> > > >
> > > > **[Performance with different evaluation functions]**
> > > >
> > > > Table B reports results when using each metric, or their weighted combinations, is used as DATE’s evaluation function. Note that PickScore values differ from those in the original rebuttal due to a bug fix in our implementation; however, overall trends remain unchanged.
> > > >
> > > > In most cases, DATE improves performance over the fixed embedding, regardless of whether a given metric is used as the evaluation function. One exception is when AS is included, where performance on other metrics often decreases, likely because AS, being independent of the text input, offers little synergy with semantic alignment metrics, as shown in Table A. Consistent with Figure 4b in the main text, the combined objective can yield higher values than using a single metric alone (as indicated by the bold numbers in Table B), demonstrating the synergistic potential of combining metrics during text embedding optimization.
> > > >
> > > > **[Table B: Results on COCO with SD v1.5 (extension of Table 1 in the main text), including AS, PS, and various combined evaluation functions. Higher and lower weights in columns 2–5 are denoted by ◎ and O, respectively; a blank indicates the metric is not used. **Bold** numbers mark cases where combined metrics outperform the target single metric alone. Note that PickScore values differ from those in the original rebuttal due to a bug fix, but trends remain unchanged.]**
> > > > ||AS|CS|IR|PS||FID↓|AS↑|CS↑|IR↑|PS↑|
> > > > |---|---|---|---|---|---|---|---|---|---|---|
> > > > |Fixed (50 steps)||||||18.66|5.38|0.3204|0.2132|21.51|
> > > > |Fixed (70 steps)||||||18.27|5.37|0.3199|0.2137|21.50|
> > > > |**DATE (50 steps, 10% update)**|||||||||||
> > > > |with a single evaluation function|||||||||||
> > > > |AS|○|||||18.82|5.58|0.3169|0.1910|21.46|
> > > > |CS||○||||17.90|5.35|0.3237|0.2364|21.53|
> > > > |IR|||○|||18.61|5.40|0.3224|0.4792|21.53|
> > > > |PS||||○||18.49|5.42|0.3225|0.2745|21.93|
> > > > |with two combined evaluation functions|||||||||||
> > > > |AS+CS|◎|○||||18.77|5.58|0.3171|0.1911|21.46|
> > > > ||○|◎||||18.15|5.38|0.3219|0.2179|21.43|
> > > > |AS+IR|◎||○|||18.90|5.57|0.3176|0.2428|21.48|
> > > > ||○||◎|||18.15|5.43|0.3216|0.4575|21.53|
> > > > |AS+PS|◎|||○||18.81|5.58|0.3175|0.2091|21.54|
> > > > ||○|||◎||18.67|5.44|0.3219|0.2705|21.91|
> > > > |CS+IR||◎|○|||17.94|5.39|**0.3241**|0.4430|21.52|
> > > > |||○|◎|||18.04|5.40|0.3225|0.4756|21.53|
> > > > |CS+PS||◎||○||18.33|5.40|**0.3241**|0.2839|21.87|
> > > > |||○||◎||18.61|5.42|0.3224|0.2753|21.93|
> > > > |IR+PS|||◎|○||18.15|5.41|0.3226|0.4774|21.63|
> > > > ||||○|◎||18.54|5.42|0.3227|0.3126|21.93|
> > > > |with three combined evaluation functions|||||||||||
> > > > |AS+CS+IR|◎|○|○|||18.87|5.57|0.3179|0.2487|21.48|
> > > > ||○|◎|○|||18.50|5.46|0.3208|0.3332|21.50|
> > > > ||○|○|◎|||18.07|5.43|0.3219|0.4557|21.53|
> > > > |AS+CS+PS|◎|○||○||18.96|5.57|0.3179|0.2092|21.55|
> > > > ||○|◎||○||18.62|5.47|0.3211|0.2450|21.69|
> > > > ||○|○||◎||18.69|5.44|0.3221|0.2705|21.91|
> > > > |AS+IR+PS|◎||○|○||18.93|5.57|0.3186|0.2602|21.56|
> > > > ||○||◎|○||18.33|5.44|0.3222|0.4599|21.61|
> > > > ||○||○|◎||18.72|5.44|0.3224|0.3071|21.91|
> > > > |CS+IR+PS||◎|○|○||18.32|5.41|**0.3244**|0.4192|21.81|
> > > > |||○|◎|○||18.16|5.41|0.3229|**0.4812**|21.63|
> > > > |||○|○|◎||18.68|5.42|0.3228|0.3138|21.92|

---

> > > > > ### Comment · Reviewer_dcFc · 2025-08-08
> > > > >
> > > > > Thank you for the detailed analysis of the reward models. My concerns are addressed well. I will recommend to accept this paper.

---

> > > > > > ### Author Response · Authors · 2025-08-08
> > > > > >
> > > > > > Thank you for the response. We're glad to hear that the concerns are addressed well, and we truly appreciate the time and effort dedicated to the review.

---

### Official Review · Reviewer_nC2W · 2025-07-01

**Clarity:** 3
**Significance:** 3
**Originality:** 3
**Rating:** 4
**Confidence:** 4

**Summary:**

This paper proposes Diffusion Adaptive Text Embedding (DATE), a method for dynamically updating text embeddings at each diffusion sampling timestep in text-to-image diffusion models. Existing methods typically use a fixed text embedding throughout the reverse diffusion process, which may lead to suboptimal semantic alignment. DATE refines the text embedding per timestep based on intermediate perturbed images by optimizing a differentiable image-text alignment score (e.g., CLIP score or ImageReward). The paper formulates this as a constrained optimization problem and proposes a lightweight gradient-based update rule that does not require additional model training. DATE is model-agnostic, efficient (can be applied at test time), and improves performance on tasks such as multi-concept generation and text-guided image editing.

**Questions:**

1. How well does DATE generalize to long and complex prompts? Are there cases where it introduces instability or performance degradation?
2. Does the stronger text-image alignment induced by DATE reduce image diversity or cause overfitting to the reward function?
3. Can DATE be effectively combined with prompt tuning or soft prompt methods? Would such combinations be complementary or interfere with each other?

**Ethical Concerns:**

["NO or VERY MINOR ethics concerns only"]

**Final Justification:**

I have read the authors' rebuttal. Although some of the explanations and experiments provided are not sufficiently thorough, after considering the rebuttal and the feedback from other reviewers, I believe the work still meets the acceptance criteria. I appreciate the additional experiments and analyses, but they do not fully resolve the main weaknesses. Therefore, I will maintain my original score.

**Limitations:**

Yes

**Quality:**

4

**Strengths And Weaknesses:**

Strengths:

* The idea of introducing time-varying text embedding updates during the diffusion sampling process is both innovative and intuitive Furthermore, optimizing the alignment score at intermediate steps of the diffusion process, rather than only at the final output, is a novel training strategy. The method is accompanied by theoretical derivations.
* The paper presents well-designed experiments and extensive ablation studies.

Weaknesses:

* Although the authors employ strategies such as sparse updates to mitigate runtime, DATE still incurs a noticeable overhead compared to standard diffusion sampling (over 30% inference time). This may limit its applicability in resource-constrained settings.

* The effectiveness of DATE is dependent on the choice of reward function (CLIP, ImageReward). There exists a risk of overfitting to proxy metrics, which may not fully reflect human judgment or semantic correctness.

---

> ### Author Rebuttal · Authors · 2025-07-31
>
> We thank the reviewer for the valuable feedback. Below are our responses to the concerns.
> ***
> > ### **W1: [Computational overhead]**
>
> The increased GPU memory usage in DATE mainly stems from gradient computations, as noted in the Limitations section. To address this, we apply half-precision (FP16) inference during sampling. As shown below, this reduces runtime and memory with only a slight performance drop. Still, DATE outperforms fixed embedding.
>
> **[Table: Results on COCO with SD v1.5  of Table 1 in the main text, including FP16.]**
>
> ||Time|Memory|FID↓|CLIP score↑|ImageReward↑|
> |---|---|---|---|---|---|
> |Fixed (50 steps)|5.64|24.0|18.66|0.3204|0.2132|
> |Fixed (70 steps)|7.87|24.0|18.27|0.3199|0.2137|
> |**DATE** (50 steps, 10% update)|
> |&nbsp;with CLIP score|7.82|61.5|17.90|0.3237|0.2364|
> |&nbsp;with CLIP score (FP16)|4.40|32.9|17.99|0.3229|0.2265|
> |&nbsp;with ImageReward|7.82|61.5|18.61|0.3224|0.4792|
> |&nbsp;with ImageReward (FP16)|4.02|30.6|18.03|0.3222|0.4773|
>
> One important note is that in our implementation, casting the CLIP model used in the evaluation function to FP16 resulted in degraded performance. Therefore, we kept this component in full precision. Nevertheless, since the diffusion model accounts for most of the computational overhead, applying FP16 to the rest of the pipeline still provides substantial savings.
>
> It is also worth noting that FP16 is a general memory-efficient strategy that can be applied across different sampling methods. In this context, our results show that DATE remains compatible with such strategies, despite its gradient computations. However, we do not claim that FP16 integration represents a fundamental improvement to DATE itself in terms of memory or speed. Rather, it shows that DATE can effectively incorporate standard efficiency techniques without loss of compatibility.
> ***
> > ### **W2: [Dependency on reward function]**
>
> We acknowledge that DATE depends on the reward function, as discussed in Appendix E. Here, we offer theoretical, empirical, and practical perspectives.
>
> **(Theoretical perspective)**
>
> Theorem 2 shows DATE aligns text embeddings with the reward function while also regularizing with the model's text-conditional likelihood. This means that DATE does not blindly overfit to the reward function but instead applies regularized guidance informed by the underlying diffusion model. This effect can reduce the risk of overfitting to the proxy metrics.
>
> **(Empirical perspective)**
>
> Empirically, as shown in Table 1 of the main text, DATE improves performance on both the reward function used for optimization and other metrics not used during optimization (e.g., FID). To further validate this generalization, we conduct additional experiments using PickScore (Kirstain et al., 2023) as both an optimization objective and an external evaluation metric. As shown in the table, DATE consistently improves performance across all metrics. This suggests that DATE’s improvements reflect broader enhancements in generation quality. Ongoing research continues to explore more faithful evaluation metrics for text-to-image generation, and DATE can be readily combined with such advances.
>
> **[Table: Results on COCO with SD v1.5 of Table 1 in the main text, including PickScore.]**
> ||Time|FID↓|CLIP score↑|ImageReward↑|PickScore↑|
> |---|---|---|---|---|---|
> |Fixed (50 steps)|5.64|18.66|0.3204|0.2132|16.8590|
> |Fixed (70 steps)|7.87|18.27|0.3199|0.2137|16.8646|
> |**DATE** (50 steps, 10% update)|
> |&nbsp;with CLIP score|7.82|**17.90**|**0.3237**|0.2364|16.8728|
> |&nbsp;with ImageReward|7.82|18.61|0.3224| **0.4792**|16.8750|
> |&nbsp;with PickScore|8.23|18.49|0.3225|0.2745|**16.8761**|
>
> (Kirstain et al., 2023) Pick-a-Pic: An Open Dataset of User Preferences for Text-to-Image Generation, NeurIPS 2023.
>
> **(Practical perspective)**
>
> Our method also supports multi-objective optimization, which can further mitigate overfitting to a single metric. In Figure 4 of the main text, we show results using a combined objective of CLIP score and ImageReward. Interestingly, this combination outperforms the use of each metric individually. This result highlights the potential of our method to leverage synergy between multiple objectives for more robust guidance.
>
> In summary, while reward function dependency is an important consideration, DATE incorporates both theoretical regularization and practical flexibility to address it effectively.
> ***
> > ### **Q1: [Long and complex prompts]**
>
> Text encoders in diffusion models are often limited by a fixed token length, which can pose challenges for handling long and complex prompts. For example, many diffusion models using CLIP encoders are restricted to 77 tokens, and the tokens beyond this limit are typically truncated and ignored. Since DATE provides guidance through an external function and does not rely solely on the internal text encoder. This allows DATE to influence the generation process beyond the model’s fixed token-length limitation, making it a promising solution for handling long and complex prompts.
>
> To investigate this, we perform experiments on the LongAlign (Liu et al., 2025), which contains prompts of substantial length and complexity. We sample 300 prompts and use ImageReward (20% updates) as the evaluation function. For ImageReward, each long prompt is split into individual sentences, and at each update step, one sentence is randomly selected to compute the score and guide the embedding update. We use Denscore (Liu et al., 2025) for the metric.
>
> **[Table: Results for long prompts on LongAlign dataset (SD v1.5)]**
> |Method|Denscore↑|
> |---|---|
> |Fixed|0.2034|
> |**DATE**|**0.2042**|
>
> **[Table: Average Denscore by sentence position]**
> ||S1|S2|S3|S4|S5|S6+|
> |---|---|---|---|---|---|---|
> |Fixed|**0.2597**|**0.2478**|**0.2234**|0.1961|0.1678|0.1764|
> |**DATE**|0.2561|0.2465|0.2230|**0.1991**|**0.1729**|**0.1799**|
>
> These results show that DATE improves Denscore compared to the baseline. Notably, the improvement is more significant for sentences appearing later in the prompt, suggesting that DATE can inject additional semantic guidance in regions where the base model typically underperforms due to token truncation. These results suggest that DATE can serve as an effective mechanism for injecting auxiliary information into diffusion models, especially in scenarios where prompt length exceeds the native capacity of the text encoder.
>
> (Liu et al., 2025) Improving Long-Text Alignment for Text-to-Image Diffusion Models, ICLR 2025.
> ***
> > ### **Q2: [Diversity]**
>
> As discussed in our response to W2, our theoretical analysis based on Theorem 2 suggests that DATE is regularized by the underlying diffusion model, which helps mitigate overfitting to the reward function. Additionally, since DATE does not modify model weights and instead performs instance-specific updates during inference, we believe its impact on sample diversity is limited.
>
> To empirically validate this, we measure diversity using Recall, the metric for evaluating the diversity of generated images in generative models.
>
> **[Table: Results on the COCO from Table 1 in the main paper with SD v1.5 including Recall.]**
> |Method|FID↓|CLIP score↑|ImageReward↑|Recall↑|
> |---|---|---|---|---|
> |Fixed (50 steps)|18.66|0.3204|0.2132|0.8132|
> |Fixed (70 steps)|18.27|0.3199|0.2137|0.8154|
> |EBCA|25.85|0.2877|-0.3128|0.6960|
> |Universal Guidance|18.56|0.3216|0.2221|0.8048|
> |**DATE** (50 steps)|
> |&nbsp;10% update with CLIP score|17.90|0.3237|0.2364|0.8178|
> |&nbsp;all updates with CLIP score|**17.22**|**0.3292**|0.2277|0.8208|
> |&nbsp;10% update with ImageReward|18.61|0.3224|0.4792|0.8225|
> |&nbsp;all updates with ImageReward|18.17|0.3224|**1.2972**|**0.8558**|
>
> As shown in the results, DATE improves both text-image alignment and sample diversity. We speculate that giving the model the opportunity to dynamically update the text embedding across timesteps, rather than relying on a fixed embedding, may have encouraged greater exploration during sampling. However, the underlying mechanism behind this effect warrants further investigation.
> ***
> > ### **Q3: [Combined with prompt tuning]**
>
> Our method can be combined with prompt tuning and soft prompt methods. Several learning-based approaches (Hao et al., 2023, Mo et al., 2024) have explored prompt optimization using reward models and reinforcement learning. While such approaches focus on optimizing prompt tokens or prompt-generating models, DATE operates at inference time by directly adjusting the text embeddings without any additional training. Therefore, they can be used together: refined prompts from these methods can serve as the initial input, and DATE can further refine the embedding during sampling to better align with the evaluation function.
>
> We conduct additional experiments by combining DATE with Promptist (Hao et al., 2023), a representative prompt tuning method. We use ImageReward as the evaluation function for DATE.
>
> **[Table: Results with the prompt tuning, Promptist (Hao et al., 2023). FID values may differ in scale due to the smaller sample size (1k).]**
> |Method| FID-1K↓ |CLIP score↑ | ImageReward↑ | Aesthetic score↑ |
> |---|---|---|---|---|
> |Fixed|48.81|0.3206|0.1801|5.40|
> |**DATE**|48.09|0.3233|0.4680|5.43|
> |Promptist|57.75|0.2785|-0.0380|6.23|
> |Promptist+**DATE**|56.23|0.2827|0.2471|6.20|
>
> Promptist focuses on improving aesthetic quality (represented by Aesthetic score), often at the cost of other metrics like CLIP score, FID, and ImageReward. In contrast, DATE improves all metrics, including Aesthetic score, which is not used in its objective, indicating good generalization. When combined with Promptist, DATE helps recover image-text alignment and overall quality while maintaining high aesthetic quality. This suggests DATE can complement prompt tuning methods.
>
> (Hao et al., 2023) Optimizing Prompts for Text-to-Image Generation, NeurIPS 2023.
> (Mo et al., 2024) Dynamic Prompt Optimizing for Text-to-Image Generation, CVPR 2024.

---

> > ### Comment · Reviewer_nC2W · 2025-08-05
> > **Reviewer Feedback on the Rebuttal**
> >
> > The reviewers share concerns about handling complex prompts, the close link between metric gains and the reward function (suggesting overfitting), limited comparison to prompt-tuning methods, and extra computational cost. The authors’ long-text prompt benchmark shows only marginal improvement, and the compared baseline, Promptist, performs poorly and thus offers limited persuasive power. Despite these issues, after considering the rebuttal and other reviewers’ feedback, I believe the work still meets the acceptance criteria. I appreciate the added experiments and analyses, but they do not fully address the main weaknesses, so I will maintain my original score.

---

> > > ### Author Response · Authors · 2025-08-08
> > >
> > > Thank you for the constructive feedback and for bringing up the remaining concerns regarding the complex prompt and the prompt-tuning baseline. Regarding the compared baseline, we acknowledge that the baseline has relatively low semantic alignment performance. We will try combining DATE with a stronger prompt-tuning method, and we will include the results in a revised version.
> > >
> > > In the rebuttal, our long-text prompt benchmark was presented as a new application of DATE for handling extremely long text inputs that exceed the token limit of the text encoder. In addition, we have now evaluated DATE on GenAI-Bench (Lin et al., 2024), a recently introduced challenging benchmark designed to assess text-to-image generative models under complex and compositional text prompts, including multiple objects, attribute bindings, spatial/action relations, counting, and logical reasoning. We also report the VQAScore metrics proposed by Lin et al. (2024).
> > >
> > > As shown in the below table, DATE improves all metrics over the fixed embedding baseline in these complex prompts. Furthermore, since both methods generate images from the same random seeds, we conducted paired t-tests to assess statistical significance. Except for the ImageReward value of DATE with CLIP score (p=0.0176, still statistically significant at the 5% level), all improvements had p-values below 0.0001. These results indicate that DATE remains effective even for highly complex prompts.
> > >
> > > **[Table: Results on GenAI-Bench (Lin et al., 2024) for complex prompts, comparing DATE with fixed embeddings.]**
> > > | Method | CLIP score↑ | ImageReward↑ | VQAScore↑ |
> > > |---|---|---|---|
> > > | Fixed | 0.3198 | 0.1631 | 0.6282 |
> > > ||
> > > | **DATE** with CLIP score (10% update) | **0.3226** | 0.1774 | **0.6389** |
> > > | **DATE** with ImageReward (10% update) | 0.3212 | **0.4033** | 0.6376 |
> > >
> > > We sincerely appreciate the time and effort you dedicated to reviewing our work and considering these additional results.
> > >
> > > ***
> > >
> > > (Lin et al, 2024) Evaluating Text-to-Visual Generation with Image-to-Text Generation, ECCV 2024.

---

### Official Review · Reviewer_1BqT · 2025-07-03

**Clarity:** 3
**Significance:** 3
**Originality:** 3
**Rating:** 5
**Confidence:** 4

**Summary:**

This paper introduces Diffusion Adaptive Text Embedding (DATE), a framework for improving text-to-image diffusion models by dynamically updating the text embedding at each diffusion timestep. Unlike traditional models that use a fixed text embedding across all timesteps, DATE adapts the embedding based on the current denoised image, enhancing semantic alignment. The update process is guided by maximizing a text-image alignment metric using a gradient-based optimization. Theoretical analysis supports that DATE improves alignment without degrading generative quality. Experiments show consistent performance gains across tasks such as multi-concept generation and text-guided image editing.

**Questions:**

Q1. The authors used two metrics (CLIPScore and ImageReward) for optimization. Can another metric can be applied for optimization?

Q2. For further questions and concerns, please refer to the Major and Minor Weaknesses sections above.

**Ethical Concerns:**

["NO or VERY MINOR ethics concerns only"]

**Final Justification:**

During the rebuttal, my major concerns regarding (1) comparison with additional baselines, (2) novelty, and (3) computational cost were resolved. Therefore, I have increased my score to 5.

**Limitations:**

yes

**Quality:**

2

**Strengths And Weaknesses:**

**Strengths**

S1. This paper presents a clearly motivated framework for text-to-image generation using diffusion models. Guiding the diffusion model with adaptive text embeddings is both intuitive and effective, since it does not require the training of the diffusion model parameters.

S2. The proposed method is based on strong mathematical formulation (Sec. 3.1. ~ 3.2.), supported by theoretical analysis (Sec. 3.3., Appendix A), which makes the proposed method solid.

S3. The experimental results (including supplementary materials) are impressive. DATE outperforms baselines in FID [1], CLIP Score [2] and ImageReward [3]. Notably, optimizing the method with a specific metric (e.g., CLIP Score [2]) still leads to improvements across other evaluation metrics, including FID [1] and ImageReward [3].

S4. The proposed method demonstrates good applicability and generalizability. DATE can be applied for both text-guided image editing and multi-concept generation, and can be incorporated into orthogonal prior work (e.g. DDPM Inversion [4]), further extending its versatility.

**Major Weakness**

W1. The comparison with prior work is limited. Baselines including prompt optimization-based methods [5, 6, 7], reward-based methods [8], and other guidance strategies [9, 10] should be involved for a more comprehensive and strong evaluation. Is optimizing the text embedding truly the most effective approach for improving text-to-image diffusion models, compared to other techniques?

W2. Along with W1, the paper should better clarify its novelty compared to prior prompt optimization-based works, especially with regard to what DATE offers beyond existing strategies.


**Minor Weakness**

W3. The paper would benefit from visualizing diverse qualitative results from both the baseline algorithms and the proposed method for a more fair comparison.

W4. DATE requires x2.5 GPU memory for a batch size of 4 (Table 8) compared to the method using fixed text embedding. Is there a lightweight method (revised version) that can reduce the GPU memory usage?

References

[1] Heusel, Martin, et al. "Gans trained by a two time-scale update rule converge to a local nash equilibrium." in NeurIPS (2017).

[2] Hessel, Jack, et al. "Clipscore: A reference-free evaluation metric for image captioning.” in EMNLP (2021).

[3] Xu, Jiazheng, et al. "Imagereward: Learning and evaluating human preferences for text-to-image generation." in NeurIPS (2023).

[4] Huberman-Spiegelglas, Inbar, Vladimir Kulikov, and Tomer Michaeli. "An edit friendly ddpm noise space: Inversion and manipulations." in CVPR (2024).

[5] Mo, Wenyi, et al. "Dynamic prompt optimizing for text-to-image generation." in CVPR (2024).

[6] Um, Soobin, and Jong Chul Ye. "Minority-Focused Text-to-Image Generation via Prompt Optimization." in CVPR (2025).

[7] Hao, Yaru, et al. "Optimizing prompts for text-to-image generation." in NeurIPS (2023).

[8] Kim, Semin, et al. "Reward-Agnostic Prompt Optimization for Text-to-Image Diffusion Models." arXiv preprint (2025).

[9] Sadat, Seyedmorteza, et al. "CADS: Unleashing the diversity of diffusion models through condition-annealed sampling." in ICLR (2024).

[10] Um, Soobin, and Jong Chul Ye. "Self-guided generation of minority samples using diffusion models." in ECCV (2024).

---

> ### Author Rebuttal · Authors · 2025-07-31
>
> We appreciate the thorough reviews and valuable comments. We address the concerns below.
> ***
> > ### **W1 & W2: [Previous works]**
>
> Thank you for pointing out these relevant lines of work. We agree that a more comprehensive comparison with prompt optimization and guidance strategies would strengthen the paper. In the revised version, we include both a detailed discussion and additional experiments covering the following areas, as below.
>
> **(Prompt optimization method)**
>
> Several learning-based approaches [5, 7] generate static prompts via training, often with reward models or reinforcement learning. These methods are effective but costly, and fixed prompts lack adaptability during sampling. In contrast, DATE is training-free and dynamically updates text embeddings at inference time across timesteps. This enables fine-grained control and responsiveness to intermediate generations.
>
> [6] is a concurrent method that also performs inference-time prompt optimization by updating a placeholder token for minority instance generation. While [6] focuses on a specific task and optimizes a single token, DATE is a general framework that updates the full embedding vector and is applicable across tasks.
> [8] proposes a training-free, gradient-free method that leverages two LLMs to iteratively refine prompts. While this method avoids training, it requires multiple LLM inferences and image generations per prompt, making it computationally expensive. In contrast, DATE uses a single image trajectory per instance and updates the text embedding sequentially at each timestep (Motivation 1 from Section 3.1 in the main paper).
>
> We evaluate DATE individually and in combination with Promptist [7] (trained prompts for aesthetics) using SD v1.4 and ImageReward:
>
> **[Table: Results with the prompt tuning method, Promptist [7]. Note: FID values may differ in scale from those in the main paper due to the smaller sample size (1k).]**
>
> | Method             | FID-1K↓ | CLIP score↑ | ImageReward↑ | Aesthetic score↑ |
> |--------------------|---------|-------------|--------------|------------------|
> | Fixed    | 48.81   | 0.3206      | 0.1801       | 5.40             |
> | **DATE**             | 48.09   | 0.3233      | 0.4680       | 5.43             |
> | Promptist        | 57.75   | 0.2785      | -0.0380      | 6.23             |
> | Promptist + **DATE** | 56.23   | 0.2827      | 0.2471       | 6.20             |
>
> Promptist focuses on improving aesthetic quality (represented by Aesthetic score), often at the cost of other metrics like CLIP score, FID, and ImageReward. In contrast, DATE improves all metrics, including Aesthetic score, which is not used in its objective, indicating good generalization. When combined with Promptist, DATE helps recover image-text alignment and overall quality while maintaining high aesthetic quality. This suggests DATE can complement prompt tuning methods by improving balance and robustness during inference.
>
> **(Guidance strategies)**
>
> As discussed in the main paper, Universal Guidance (UG) applies external functions to guide sampling in the perturbed data space. [10] targets minority sample generation by proposing a metric that can be internally computed within the diffusion model and applying guidance in the data space.
>
> In contrast, DATE applies guidance in the text embedding space, which introduces a fundamental distinction from data-space guidance methods. In Section 4.3, Theorem 2 provides a theoretical foundation showing that DATE updates the text embedding in a direction that aligns with the evaluation function while simultaneously accounting for the model’s original text-conditional likelihood. This enables DATE to strike a better balance between semantic alignment and preservation of the diffusion model’s distribution. Experimentally, we also show in Table 1 that DATE outperforms UG.
>
> Meanwhile, CADS [9] shows that injecting time-dependent Gaussian noise into the conditional embedding can improve generation diversity. We empirically compare DATE with CADS and evaluate their combination. The experiments follow the same setup as Table 1 in the main paper, with DATE using 10% update with ImageReward configuration. Given CADS’s known effectiveness in improving diversity, we include Recall as an additional diversity metric.
>
> **[Table: Performance on the COCO validation set with CADS [9], including Recall metric.]**
>
> | Method      | FID-5K↓   | CLIP score↑ | ImageReward↑ | Recall↑    |
> |-------------|-----------|-------------|--------------|------------|
> | Fixed       | 18.66     | 0.3204      | 0.2132       | 0.8132     |
> | **DATE**        | 18.61     | **0.3224**  | **0.4792**   | 0.8225     |
> | CADS        | 16.62     | 0.3119      | -0.0534      | 0.8328     |
> | CADS + **DATE** | **16.52** | 0.3122      | 0.1973       | **0.8434** |
>
> As shown in the table, CADS (third row) improves diversity, as reflected by better FID and Recall, but at the cost of reduced text-image alignment, with lower CLIP score and ImageReward. In contrast, DATE (second row) improves both alignment and diversity metrics. When CADS and DATE are combined (last row), the result shows a synergistic effect: FID and Recall improve further, while alignment metrics also increase relative to CADS alone. This suggests that while CADS excels in enhancing diversity, it does not improve text-image alignment, which can be effectively complemented by combining it with DATE.
>
> ***
>
> > ### **W3: [Qualitative results]**
>
> Thank you for the suggestion. We currently provide qualitative comparisons for our application tasks in the main text (Figures 11 and 12) and in the appendix (Figures 17 and 18). The review system does not allow uploading additional images during the rebuttal period. In our revision, we will add diverse generated samples, e.g., COCO-prompt images corresponding to Table 1, to provide clearer visual comparisons.
>
> ***
>
> > ### **W4: [Lightweight method]**
>
> The increase in GPU memory usage for DATE is primarily due to the gradient computations, as noted in the Limitations section. To address this, we explore memory-efficient techniques. In particular, we apply half-precision (FP16) inference during the sampling process. As shown in the below table, this reduces runtime and memory consumption, albeit at the cost of slightly decreased performance. Nevertheless, our methods still outperform fixed embedding methods.
>
> **[Table: Computational resources and performance on the COCO validation set. This is an extended version of Table 1 in the main text, including results with half-precision.]**
>
> |                             | Time | Memory | FID↓  | CLIP score↑ | ImageReward↑ |
> |-----------------------------|------|--------|-------|-------------|--------------|
> | Fixed (50 steps)            | 5.64 | 24.0   | 18.66 | 0.3204      | 0.2132       |
> | Fixed (70 steps)            | 7.87 | 24.0   | 18.27 | 0.3199      | 0.2137       |
> | **DATE** (50 steps, 10% update) |      |        |       |             |              |
> | &nbsp;with CLIP score         | 7.82 | 61.5   | 17.90 | 0.3237      | 0.2364       |
> | &nbsp;with CLIP score (FP16)  | 4.40 | 32.9   | 17.99 | 0.3229      | 0.2265       |
> | &nbsp;with ImageReward        | 7.82 | 61.5   | 18.61 | 0.3224      | 0.4792       |
> | &nbsp;with ImageReward (FP16) | 4.02 | 30.6   | 18.03 | 0.3222      | 0.4773       |
>
> One important note is that in our implementation, casting the CLIP model used in the evaluation function to FP16 resulted in degraded performance. Therefore, we kept this component in full precision. Nevertheless, since the diffusion model accounts for most of the computational overhead, applying FP16 to the rest of the pipeline still provides substantial savings.
>
> It is also worth noting that FP16 is a general memory-efficient strategy that can be applied across different sampling methods. In this context, our results demonstrate that DATE remains compatible with such strategies, despite its additional gradient computations. However, we do not claim that FP16 integration represents a fundamental improvement to DATE itself in terms of memory or speed. Rather, it shows that DATE can effectively incorporate standard efficiency techniques without loss of compatibility.
>
> ***
>
> > ### **Q1: [Another metric for optimization]**
>
> Our method is compatible with any evaluation metric that provides access to model weights. We experiment with Pickscore (Kirstain et al., 2023), another popular text-to-image evaluation metric, as the optimization objective. Similar to our results with CLIPScore and ImageReward, using PickScore also led to consistent improvements across all evaluation metrics. Additionally, when evaluating images optimized with other metrics using PickScore, our method still outperformed the fixed embedding baseline.
>
> **[Table: Performance on the COCO validation set using Stable Diffusion v1.5 with DDIM sampler. This is an extended version of Table 1 in the main text, including results with PickScore.]**
> |                             | Time | FID↓      | CLIP score↑ | ImageReward↑ | PickScore↑  |
> |-----------------------------|------|-----------|-------------|--------------|-------------|
> | Fixed (50 steps)            | 5.64 | 18.66     | 0.3204      | 0.2132       | 16.8590     |
> | Fixed (70 steps)            | 7.87 | 18.27     | 0.3199      | 0.2137       | 16.8646     |
> | **DATE** (50 steps, 10% update) |      |           |             |              |             |
> | &nbsp;with CLIP score         | 7.82 | **17.90** | **0.3237**  | 0.2364       | 16.8728     |
> | &nbsp;with ImageReward        | 7.82 | 18.61     | 0.3224      | **0.4792**   | 16.8750     |
> | &nbsp;with PickScore          | 8.23 | 18.49     | 0.3225      | 0.2745       | **16.8761** |
>
> (Kirstain et al., 2023) Pick-a-Pic: An Open Dataset of User Preferences for Text-to-Image Generation, NeurIPS 2023.

---

> > ### Comment · Reviewer_1BqT · 2025-08-03
> > **Thanks for the rebuttal**
> >
> > I appreciate the authors’ effort during the rebuttal process. My concerns have been fully addressed, especially in the following aspects:
> >
> > **W1 & W2.** The authors compared DATE with additional baselines (Promptist, CADS) as well as their combinations with DATE. As shown in the table, DATE consistently outperforms the baselines, demonstrating the effectiveness of the proposed methods.
> >
> > **W4**. A lightweight version has been introduced, which improves computational efficiency.
> >
> > **Q1.** The authors claimed that any evaluation metric that provides model weights can be used, and supported this claim with experiments using Pickscore. I appreciate these additional experiments, which show the generability of DATE.
> >
> > In conclusion, considering all of the above, I’ve decided to increase my score from 4 to 5.

---

> > > ### Author Response · Authors · 2025-08-03
> > >
> > > Thank you for your thoughtful feedback and for updating your score. We're glad our responses addressed your concerns. We appreciate your time and support.

---

### Note · Authors · 2025-08-12

We greatly appreciate the reviewers’ comprehensive reviews and insightful suggestions.

We are encouraged by the consistent recognition of the proposed method, DATE, as a clearly motivated and novel framework in which guiding the text embedding is intuitive and effective. Reviewers also highlighted its solid theoretical analysis, clear mathematical formulation, and impressive experimental results.

We addressed the main concerns through the following clarifications and additional results.

* **Reliance on the evaluation function** (1BqT Q1; nC2W W2; dcFc W1, additional comment; TJGJ W3)
  * DATE with PickScore showed consistent improvements. We also explored multi-metric objectives (combining Aesthetic score, CLIP score, ImageReward, and PickScore) to jointly consider image fidelity, semantic alignment, and human preference. These combinations often yielded further gains, showing DATE’s robustness across diverse objectives.

* **Comparison with previous work** (1BqT W1 & W2; nC2W Q3; dcFc W3)
  * We added a discussion of prompt-tuning, reward-based fine-tuning, and diffusion guidance methods. Furthermore, we empirically compared with the prompt-tuning method (Promptist) and the guidance method (CADS), showing DATE consistently outperforms and complements these methods.

* **Computational overhead** (1BqT W4, nC2W W1)
  * We introduced an FP16 variant that improves efficiency with minimal performance drop.

* **Long and complex prompts** (nC2W Q1, additional comment; TJGJ W1)
  * We proposed an approach to incorporate truncated tokens for over-length prompts, empirically validated it on LongAlign. We also achieved statistically significant gains on GenAI-Bench, a complex and compositional prompt set.

* **Application to recent models** (dcFc W2; TJGJ W2 & Q)
  * We demonstrated that DATE can effectively be applied to the recent stronger architectures with multiple text embeddings, including SD3, FLUX, and SDXL.

We believe these results help alleviate the reviewers’ concerns. We once again thank the reviewers for their constructive feedback and will incorporate these clarifications, analyses, and additional results into the revised version.

---

### Decision · Program_Chairs · 2025-09-17

**Decision:**

Accept (poster)

**Comment:**

The authors present Diffusion Adaptive Text Embedding (DATE), which learns text embeddings in text-to-image diffusion, unlike previous methods that kept embeddings frozen. The experimental results encompass several tasks, including text-to-image generation, multi-concept generation, and text-guided image editing.

**Weaknesses:** Reviewers raised concerns about the ablations and evaluations. For instance, performance on complex prompts showed only marginal improvements in the authors' long-text benchmark. Comparisons to prompt-tuning methods were limited, as the chosen baseline (Promptist) was weak.  Additional concerns included the extra computational cost.

**Recommendation:** Accept (poster). Despite these drawbacks, all reviewers moved their scores to accept, agreeing on the importance of the technique to the community. I concur that studies probing optimization in pre-trained representations are valuable. I see no reason to intervene in the reviewers' decision.

**Discussion summary:** **1BqT**: Asked for more baselines, novelty, visuals, and efficiency. Authors added comparisons with Promptist & CADS, a FP16 lightweight variant, and PickScore generalization. Concerns resolved. **nC2W**: Flagged overhead, reward-function dependence, and diversity risks. Authors demonstrated FP16 gains, performed theoretical/empirical checks, presented long-text & GenAI-Bench results, Explored Recall diversity, and discussed prompt-tuning combinations. Some concerns remain. Score unchanged.   **dcFc**: Concerned about reward bias, missing related work, and intent deviation. The authors analyzed multiple reward models, added correlation studies, discussed SD3/multi-embedding, and showed theoretical safeguards. Rebuttal resolved issues. Score raised.   **TJGJ**: Wanted strong baselines (GPT-4o, FLUX), complex prompts, and metric justification. Authors applied DATE to SDXL/FLUX/SD3, ran long-text + complex benchmarks, and showed metric consistency. Concerns addressed; score maintained.